# MODEL EXPLANATION DISPARITIES AS A FAIRNESS DIAGNOSTIC

## ABSTRACT

Recent works on fairness in machine learning have focused on quantifying and eliminating bias against protected subgroups, and extended these results to more complex subgroups beyond simple discrete classes, known as "rich subgroups." Orthogonally, recent works in model interpretability develop local feature importance methods that, given a classifier $h$ and test point $x$, attribute influence for the prediction $h(x)$ to the individual features of $x$. This raises a natural question: *Do local feature importance methods attribute different feature importance values on average in protected subgroups versus the whole population, and can we detect these disparities efficiently?* In this paper, we formally introduce the notion of feature importance disparity (`FID`) in the context of rich subgroups, which could be used as a potential indicator of bias in the model or data generation process. We design an oracle-efficient algorithm to identify large `FID` subgroups and conduct a thorough empirical analysis auditing for these subgroups across 4 datasets and 4 common feature importance methods of broad interest to the machine learning community. Our algorithm finds (feature, subgroup) pairs that: (i) have subgroup feature importance that is often an order of magnitude different than the importance on the whole dataset (ii) generalize out of sample, and (iii) yield interesting discussions about potential bias inherent in these common datasets.

## 1 INTRODUCTION

Machine learning is rapidly becoming a more important yet more opaque part of our lives and decision making – with increasingly high stakes use cases such as recidivism analysis Angwin et al. (2016), loan granting and terms Dastile et al. (2020) and child protective services Keddell (2019). One of the hopes of wide-scale ML deployment has been that those algorithms might be free of our human biases and imperfections. This hope was, unfortunately, naive. Over the last decade, an interdisciplinary body of research has shown that machine learning algorithms can be deeply biased in both subtle and direct ways Barocas et al. (2019), and has focused on developing countless techniques to produce fairer models Caton & Haas (2023). One of the primary causes of model bias is bias inherent in the training data, rather than an explicitly biased training procedure. This issue is particularly concerning in light of modern machine learning, where an initial "foundation model", like GPT-3 Brown et al. (2020) or CLIP Radford et al. (2021), is trained on a large corpus of text or image data and released publicly, which an algorithm designer then fine tunes using a much smaller dataset tailored for a specific use case. Social bias present in the foundation model's training data then have the potential to propagate into the myriad use cases built on top of it Swinger et al. (2019); Liang et al. (2021). While the threat posed by these foundation models is new, the recognition that bias in the training data could propagate into unfair decisions made by a classifier is not; in a 2015 in a New York Times interview Miller (2015), Cynthia Dwork, widely regarded as pioneer in the field of algorithmic bias, describes the problem as such: *Suppose we have a minority group in which bright students are steered toward studying math, and suppose that in the majority group bright students are steered instead toward finance. An easy way to find good students is to look for students studying finance, and if the minority is small, this simple classification scheme could find most of the bright students. But not only is it unfair to the bright students in the minority group, it is also low utility.*

Unpacking this example further, the feature `is-finance-major` is predictive in finding bright students in the population at large, but not in the minority group. Meanwhile, the feature `is-math-major` is highly predictive in the minority group, but not at all in the majority group.

While it is clear in this simple example that the differing impact of these features causes the down-stream classifier to become biased, understanding how to attribute credit for a given prediction $\hat{y} = \theta(x_{test})$ across all features in $x_{test}$ is a difficult task in practical machine learning settings. Approaches that have garnered substantial attention include local model agnostic methods like LIME Ribeiro et al. (2016), SHAP Lundberg & Lee (2017), model-specific saliency maps Simonyan et al. (2013), and example-based counterfactual explanations Molnar (2022).

As we later discuss in Section 5, disparities in feature importance do not necessarily imply that a subgroup has fairness disparity as measured by conventional metrics like equalized odds or calibration, although we show in Subsection 4.3 that this is empirically often the case. Nor does finding a subgroup with high feature disparity come with a pre-defined "fix"– the disparity could be caused by many factors including (i) true underlying differences between subgroups, or (ii) differences in measurement of the features or outcome variables across subgroups. Since the specific "fixes" are highly context dependent, our method should be viewed as a tool for *generating hypotheses* about potential sources of bias, which can then be addressed by other means. For example, in Figure 4 we find that the feature `arrested-but-with-no-charges` is highly important when predicting `two-year-recidivism` on the population as a whole, but carries almost no importance on a subgroup which is largely defined by Native-American males. This could motivate further research into if this subgroup is policed in a different way than the population as a whole. Alternatively, a technical solution may be to train a separate model for this subgroup.

While this information is clearly useful to motivate further investigations into bias, in realistic settings finding these feature subgroup disparities is not easy: it is known that while a classifier may look fair when comparing a given fairness metric across a handful of sensitive subgroups, when the notion of a sensitive subgroup is generalized to encompass combinations and interactions between sensitive features (known as *rich subgroups* Kearns et al. (2019)), large disparities can emerge. Even for simple definitions of rich subgroup such as conjunctions of binary features, the number of subgroups is exponential in the number of sensitive attributes. In real settings with many (possibly interrelated) sensitive features, auditing for feature importance disparities across all rich subgroups is a daunting task. These new methods raise an obvious question in light of the prior discussion, although one that to the best of our knowledge has not been thoroughly studied: **When applied to classifiers and datasets where bias is a concern, do these feature importance notions uncover substantial differences in feature importance across protected groups, and when our protected groups correspond to rich subgroups can they be efficiently detected?**

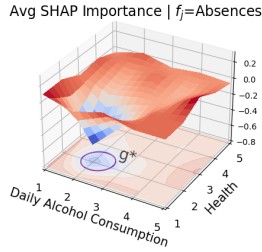

Figure 1: A toy example: On the Student dataset, the average SHAP explanatory values are substantially different for the feature `Absences` on the rich subgroup $g^*$ circled in the figure, which is defined as a function of the sensitive features `Daily Alcohol Consumption` and `Health`.

## 1.1 RESULTS

Our most important contribution is formalizing the notion of *feature importance disparity* (`FID`) in the context of feature importance notions developed in recent years, and with respect to rich subgroups (Definition 1). We categorize a feature importance notion as *separable* or not, based on whether it can be expressed as a sum over points in the subgroup (Definition 2) and define a variant of `FID`, the *average feature importance disparity* (`AVG-SEPFID`, Definition 3). Our main theoretical contribution is Theorem 1 in Section 3, which says informally that although the problem of finding the maximal `FID` subgroup is NP-hard in the worst case (Appendix E), given access to an oracle for *cost-sensitive classification* with respect to the rich subgroup class $\mathcal{G}$, (Definition 4), Algorithm 1 efficiently learns the subgroup with maximal `FID` for any separable feature importance notion.

Algorithm 1 is inspired by prior work Kearns et al. (2018); Agarwal et al. (2018); Hebert-Johnson et al. (2018) that takes a constrained optimization and solves the equivalent problem of computing the Nash equilibrium of a two-player, zero-sum game given by the Lagrangian. In Algorithm 1, we compute the equilibrium of the game by having both the min and max players implement *no-regret* strategies – which we show can be done efficiently given access to a CSC oracle for $\mathcal{G}$. This oracle is implemented via simple regression heuristic (Algorithm 2 in Appendix D), which our experimental results show works well in practice. We also show that a heuristic optimization approach works for finding high FID subgroups for a linear regression model (detailed in Appendix F).

In Section 4, we conduct a thorough empirical evaluation of our methods, auditing for large FID subgroups on the Student Cortez & Silva (2008), COMPAS Angwin et al. (2016), Bank Moro et al. (2014), and Folktables Ding et al. datasets, using LIME, SHAP, saliency maps, and linear regression coefficient as feature importance notions. Our experiments establish that:

- Across all (dataset, importance notion) pairs, we can find subgroups defined as functions of sensitive features that have large FID with respect to a given feature (Table 1, Figures 2, 3).

- Inspecting the coefficients defining the subgroups, we find interesting examples where a subgroup, defined by a few sensitive features like race and gender, has a large FID with respect to a given feature, highlighting potential bias in the data (Figures 4,5).

- Our FID's generalize out of sample and our method for controlling subgroup size is effective. (Figures 7, 8)

We also compared the maximal FID found for a rich subgroup to the FID found when we restrict our subgroup class to subgroups defined by a single sensitive attribute (marginal subgroups). In about half of the cases, the rich subgroup achieves a higher FID out of sample, justifying the use of rich subgroups (Appendix I). Taken together, our theoretical and empirical results highlight our methods as an important addition to the toolkit for detecting bias in tabular datasets with sensitive features.

## 1.2 RELATED WORK

There is substantial work investigating bias in the context of machine learning models and their training data Barocas et al. (2019); Caton & Haas (2023). We are motivated at a high level by existing work on dataset bias Kamiran & Calders (2012); Tommasi et al. (2017); Li & Vasconcelos (2019), however, to the best of our knowledge, this is the first work investigating the disparity in feature importance values in the context of rich subgroups as a fairness diagnostic. For more related work, see Appendix A.

**Anomalous Subgroup Discovery**. In terms of approach, two closely related works are Dai et al. (2022) and Balagopalan et al. (2022) which link fairness concerns on sensitive subgroups with model explanation quality, as measured by properties like stability and fidelity. Our work differs in that we are focused on the magnitude of explanation disparities themselves rather than their "quality," and that we extend our results to the rich subgroup setting. Our algorithm for searching an exponentially large subgroup space is a novel and necessary addition to work in this space. Another area of research looks to prove that a chosen score function satisfies the linear time subset scanning property Neill (2012) which can then be leveraged to search the subgroup space for classifier bias Zhang & Neill (2016); Boxer et al. (2023) in linear time. While it is hard to say with absolute certainty that this approach would not be useful it is not immediately apparent how we would force a subset scanning method to optimize over *rich subgroups*.

**Rich Subgroups and Multicalibration**. At a technical level, the most closely related papers are Kearns et al. (2018); Hebert-Johnson et al. (2018) which introduce the notion of the rich subgroup class $\mathcal{G}$ over sensitive features in the context of learning classifiers that are with respect to equalized odds or calibration. Our Algorithm 1 fits into the paradigm of "oracle-efficient" algorithms for solving constrained optimization problems introduced in Agarwal et al. (2018) and developed in the context of rich subgroups in Kearns et al. (2018; 2019); Hebert-Johnson et al. (2018). There has been much recent interest in learning multicalibrated predictors because of connections to uncertainty estimation and omnipredictors Hu et al. (2023); Gopalan et al. (2022); Jung et al. (2021). None of these works consider feature importance disparities.

**Feature Importance Notions**. For the field of interpretable or explainable machine learning, we refer to Molnar (2022) for a survey of methods. The most relevant works are methods that can be used to investigate the importance of a feature $f_j$ in a given subset of the dataset. Local explanation methods assign a feature importance for every point $(x, y)$ and define a notion of importance in a subgroup by summing or average over the points in the subgroup as we do in Definitions 2, 3.

## 2 PRELIMINARIES

Let $X^n$ represent our dataset, consisting of $n$ individuals defined by the tuple $((x, x'), y)$ where $x \in \mathcal{X}_{sense}$ is the vector of protected features, $x' \in \mathcal{X}_{safe}$ is the vector of unprotected features, and $y \in \mathcal{Y}$ denotes the label. With $X = (x, x') \in \mathcal{X} = \mathcal{X}_{sense} \times \mathcal{X}_{safe} \subset \mathbb{R}^d$ denoting a joint feature, the data points $(X, y)$ are drawn i.i.d. from a distribution $\mathcal{R}$. Let $h : \mathcal{X} \to \mathcal{Y}$ denote a classifier or regressor that predicts $y$ from $X$. We define a *rich subgroup class* $\mathcal{G} = \{g_\alpha\}_{\alpha \in \Omega}$ as a collection of functions $g : \mathcal{X}_{sens} \to [0, 1]$, where $g(x')$ denotes the membership of point $X = (x, x')$ in group $g$. Note that this is the same subgroup definition as in Kearns et al. (2018), but without the constraint that $g(x') \in \{0, 1\}$, which supports varying degrees of group membership. E.g. a biracial person may be .5 a member of one racial group and .5 a member of another. Let $f_j, j \in [d]$ denote the $j^{th}$ feature in $\mathcal{X} \subset \mathbb{R}^d$. Then for a classifier $h$ and subgroup $g \in \mathcal{G}$, let $F$ be a feature importance notion where $F(f_j, g, h)$ denote the *importance $h$ attributes to feature $j$ in the subgroup $g$*, and $F(f_j, X^n, h)$ be the importance $h$ attributes to $f_j$ on the entire dataset. We will provide more specific instantiations of $F$ shortly, but we state our definition of FID in the greatest possible generality below.

**Definition 1.** *(Feature Importance Disparity). Given a classifier $h$, a subgroup defined by $g \in G$, and a feature $f_j \in [d]$, then given a feature subgroup importance notion $F(\cdot)$, the feature importance disparity relative to $g$ is defined as:*

$$FID(f_j, g, h) = \mathbb{E}_{X \sim \mathcal{R}} |F(f_j, g, h) - F(f_j, X^n, h)|$$

We will suppress $h$ and write $FID(j, g)$ unless it is necessary to clarify what classifier we are describing. Now, given $h$ and $X^n$, our goal is to find the feature subgroup pair $(j^*, g^*) \in [d] \times \mathcal{G}$ that maximizes $FID(j, g)$, or $(j^*, g^*) = \text{argmax}_{g \in \mathcal{G}, j \in [d]} FID(j, g)$.

We now get more concrete about our feature importance notion $F(\cdot)$. First, we define the class of *separable* feature importance notions:

**Definition 2.** *(Locally Separable). A feature importance notion $F(\cdot)$ is locally separable if it can be decomposed as a point wise sum of local model explanation values $F'$:*

$$F(f_j, X^n, h) = \sum_{X \in X^n} F'(f_j, X, h)$$

It follows that for separable notions, $F(f_j, g, h) = \sum_{X \in X^n} g(X) F'(f_j, X, h)$. Given a local model explanation $F'$, we can define a more specific form of FID, the *average feature importance disparity* (AVG-SEPFID), which compares the average feature importance within a subgroup to the average importance on the dataset.

**Definition 3.** *(Average Case Locally Separable FID). For a $g \in \mathcal{G}$, let $|g| = \sum_{X \in X^n} g(X)$. Given a local model explanation $F'(\cdot)$, we define the corresponding:*

$$AVG\text{-}SEPFID(f_j, g, h) = \mathbb{E}_{X^n \sim \mathcal{R}^n} |\frac{1}{|g|} \sum_{X \in X^n} g(X) F'(f_j, X, h) - \frac{1}{n} \sum_{X \in X^n} F'(f_j, X, h)|$$

Note that AVG-SEPFID is not equivalent to a separable FID, since we divide by $|g|$, impacting every term in the summation. In Section 3, we show that we can optimize for AVG-SEPFID by optimizing a version of the FID problem with size constraints, which we can do efficiently via Algorithm 1.

This notion of *separability* is crucial to understanding the remainder of the paper. In Section 3, we show that for any *separable* FID, Algorithm 1 is an (oracle) efficient way to compute the largest FID subgroup of a specified size in polynomial time. By "oracle efficient," we follow Agarwal et al.

(2018); Kearns et al. (2018) where we mean access to an optimization oracle that can solve (possibly NP-hard) problems. While this sounds like a strong assumption, in practice we can take advantage of modern optimization algorithms that can solve hard non-convex optimization problems (e.g. training neural networks). This framework has led to the development of many practical algorithms with a strong theoretical grounding Agarwal et al. (2018); Kearns et al. (2018; 2019); Hebert-Johnson et al. (2018), and as shown in Section 4 works well in practice here as well. The type of oracle we need is called a Cost Sensitive Classification (CSC) oracle, which we define in Appendix D.

## 3 OPTIMIZING FOR AVG-SEPFID

In this section, we show how to (oracle) efficiently compute the rich subgroup that maximizes the AVG-SEPFID. Rather than optimize AVG-SEPFID directly, our Algorithm 1 solves an optimization problem that maximizes the FID subject to a group size constraint:

$$\max_{g \in \mathcal{G}} |F(f_j, g, h) - F(f_j, X^n, h)|$$

$$\text{s.t.} \quad \Phi_L(g) \equiv \alpha_L - \frac{1}{n} \sum_{X \in X^n} g(X) \leq 0, \quad \Phi_U(g) \equiv \frac{1}{n} \sum_{X \in X^n} g(X) - \alpha_U \leq 0, \tag{1}$$

where $\Phi_L$ and $\Phi_U$ are "size violation" functions given a subgroup function $g$. We denote the optimal solution to Equation 1 by $g^*_{[\alpha_L, \alpha_U]}$. We focus on optimizing the constrained FID since the following primitive also allows us to efficiently optimize AVG-SEPFID:

1. Discretize $[0, 1]$ into intervals $(\frac{i-1}{n}, \frac{i}{n}]_{i=1}^n$. Given feature $f_j$, compute $g^*_{(\frac{i-1}{n}, \frac{i}{n}]}$ for $i = 1...n$.

2. Outputting $g_{k^*}$, where $k^* = \text{argmax}_k \frac{k}{n} |F(f_j, g_k, h)|$ approximately maximizes the AVG-SEPFID given an appropriately large number of intervals $n$.

Our proof for this is available in Appendix C. We now state our main theorem, which shows that we can solve the constrained FID problem in Equation 1 with polynomially many calls to $\text{CSC}_\mathcal{G}$.

---

**Algorithm 1** Iterative Constrained Optimization

---

1: **Input:** Dataset $X^n, |X^n| = n$, hypothesis $h$, feature of interest $f_j$, separable feature importance function $F$, size constraints $\alpha_L$ and $\alpha_U$, size violation indicators $\Phi_L$ and $\Phi_U$, size penalty bound $B$, CSC oracle for $\mathcal{G}$, $\text{CSC}_\mathcal{G}(c^0, c^1)$, accuracy $\nu$.
2: **Initialize:**
3: Feature importance vector $\mathbf{C} = (F(f_j, X_i, h))_{i=1}^n$
4: Gradient weight parameter $\theta_1 = (0, 0)$
5: Learning rate $\eta = \frac{\nu}{2n^2 B}$
6: **for** $t = 1, 2, ...$ **do**
7: $\quad \lambda_{t,0} = B \frac{exp(\theta_{t,0})}{1+exp(\theta_{t,1})}, \lambda_{t,1} = B \frac{exp(\theta_{t,1})}{1+exp(\theta_{t,0})}$ $\qquad \triangleright$ Exponentiated Gradient weights
8: $\quad c_t^1 = (\mathbf{C}_i - \lambda_{t,0} + \lambda_{t,1})_{i=1}^n$
9: $\quad g_t = \text{CSC}_\mathcal{G}(\mathbf{0}, c_t^1)$ $\qquad \triangleright$ Subgroup with maximal disparity computed via CSC oracle
10: $\quad \hat{p}_\mathcal{G}^t = \frac{1}{t} \sum_{t'=1}^t g_{t'}, \lambda_t' = (B\Phi_L(\hat{p}_\mathcal{G}^t), B\Phi_U(\hat{p}_\mathcal{G}^t)), \overline{L} = L(\hat{p}_\mathcal{G}^t, \lambda_t')$
11: $\quad \hat{p}_\lambda^t = \frac{1}{t} \sum_{t'=1}^t (\lambda_{t',0}, \lambda_{t',1}), \quad g_t' = \text{CSC}_\mathcal{G}(\mathbf{0}, (\mathbf{C}_i - \hat{p}_{\lambda_0}^t + \hat{p}_{\lambda_1}^t)_{i=1}^n), \quad \underline{L} = L(g_t', \hat{p}_\lambda^t)$
12: $\quad v_t = \max\left(|L(\hat{p}_\mathcal{G}^t, \hat{p}_\lambda^t) - \underline{L}|, |\overline{L} - L(\hat{p}_\mathcal{G}^t, \hat{p}_\lambda^t)|\right)$ $\qquad \triangleright$ Check termination condition
13: $\quad$ **if** $v_t \leq v$ **then**
14: $\quad\quad$ Return $\hat{p}_\mathcal{G}^t, \hat{p}_\lambda^t$
15: $\quad$ **end if**
16: $\quad$ Set $\theta_{t+1} = \theta_t + \eta(\alpha_L - |g_t|, |g_t| - \alpha_U)$ $\qquad \triangleright$ Exponentiated Gradient update
17: **end for**

---

**Theorem 1.** *Let $F$ be a separable FID notion, fix a classifier $h$, subgroup class $\mathcal{G}$, and oracle $\text{CSC}_\mathcal{G}$. Then choosing accuracy constant $\nu$ and bound constant $B$ and fixing a feature of interest $f_j$, we will run Algorithm 1 twice; once with FID given by $F$, and once with FID given by $-F$. Let $\hat{p}_\mathcal{G}^T$ be the*

*distribution returned after $T = O(\frac{4n^2B^2}{\nu^2})$ iterations by Algorithm 1 that achieves the larger value of $\mathbb{E}[FID(j,g)]$. Then:*

$$FID(j, g_j^*) - \mathbb{E}_{g \sim \hat{p}_{\mathcal{G}}^T}[FID(j,g)] \leq \nu$$

$$|\Phi_L(g)|, |\Phi_U(g)| \leq \frac{1 + 2\nu}{B}$$

(2)

We defer the proof of Theorem 1 to Appendix B. In summary, rather than optimizing over $g \in \mathcal{G}$, we optimize over distributions $\Delta(\mathcal{G})$. This allows us to cast the optimization problem in Equation 1 as a linear program so we can form the Lagrangian $L$, which is the sum of the feature importance values and the size constraint functions weighted by the dual variables $\lambda$, and apply strong duality. We can then cast the constrained optimization as computing the Nash equilibrium of a two-player zero-sum game, and apply the classical result of Freund & Schapire (1996) which says that if both players implement *no-regret* strategies, then we converge to the Nash equilibrium at a rate given by the average regret of both players converging to zero. Algorithm 1 implements the no-regret algorithm exponentiated gradient descent Kivinen & Warmuth (1997) for the max player, who optimizes $\lambda$, and best-response via a CSC solve for the min player, who aims to maximize subgroup disparity to optimize the rich subgroup distribution.

We note that rather than computing the group $g$ that maximizes $FID(j,g)$ subject to the size constraint, our algorithm outputs a distribution over groups $\hat{p}_{\mathcal{G}}^T$ that satisfies this process *on average* over the groups. In theory, this seems like a drawback for interpretability. However, in practice we simply take the groups $g_t$ found at each round and output the ones that are in the appropriate size range, and have largest $FID$ values. The results in Section 4 validate that this heuristic choice is able to find groups that are both feasible and have large $FID$ values. This method also generalizes out of sample showing that the $FID$ is not artificially inflated by multiple testing (Appendix J). Moreover, our method provides a menu of potential groups $(g_t)_{t=1}^T$ that can be quickly evaluated for large $FID$, which can be a useful feature to find interesting biases not present in the maximal subgroup.

## 4 EXPERIMENTS

Here we report the results of our extensive empirical investigation, showing that across 16 different dataset/`FID`-notion pairings, our methods return interesting subgroups defined as a function of sensitive features that exhibit orders of magnitude `AVG-SEPFID` values that are valid out of sample. Specifically we show the following:

- Our approach works in finding subgroups defined by sensitive characteristics with high `AVG-SEPFID` across all datasets and `FID` notions studied (Figure 2).
- Examining the distribution of `AVG-SEPFID` values across the maximal subgroup for each feature, we see that they are large for a few features but tail off for the majority of features (Figure 3).
- These (subgroup, feature) pairs raise interesting questions about feature importance disparities in common datasets used in the fairness literature (Figure 4 and Appendix H).
- The discovered subgroups express disparities in fairness metrics and conversely, rich subgroups with large fairness disparities have high `AVG-SEPFID` features (Tables 2,3).

We also found that in half of the settings studied, `AVG-SEPFID` over the rich subgroup class is higher than over the marginal subgroup class, with some dataset method pairs (such as (Student, SHAP)) witnessing very large gaps (Appendix I). Algorithm 1 solves the constrained `FID` problem consistently to produce appropriately sized subgroups with sizes and `AVG-SEPFID` values that generalize to the test set across all separable `FID` notions studied (Appendix J) and that are consistent across choices of hypothesis class of $h$ (Appendix K). Our algorithms converged in a reasonable time to a (locally) optimal subgroup $g$ that maximizes the constrained `FID` (Appendix M, N).

### 4.1 EXPERIMENTAL DETAILS

**Datasets**: We used four popular datasets for the experiments: StudentCortez & Silva (2008), COMPAS Angwin et al. (2016), Bank Moro et al. (2014), and Folktables Ding et al.. For each test, we used

COMPAS twice, once predicting two-year recidivism and once predicting decile risk score (labeled COMPAS R and COMPAS D respectively in the results). For each dataset, we specified "sensitive" features which are features generally covered by equal protection or privacy laws (e.g. race, gender, age, health data). Appendix G.3 contains more details.

**Computing the `AVG-SEPFID`**: We study 3 separable notions of `FID` based on local model explanations Local-Interpretable, Model-Agnostic (LIME) Ribeiro et al. (2016), Shapley Additive Explanations (SHAP) Lundberg & Lee (2017), and the vanilla gradient approach we label GRAD Simonyan et al. (2013). For every method and dataset, we optimize the constrained `FID` over $\alpha$ ranges $(\alpha_L, \alpha_U)$ = { [.01,.05], [.05,.1], [.1,.15], [.15,.2], [.2,.25] }. These small ranges allowed us to reasonably compare the `FID` values, reported in Table 1. Additionally, these ranges span subgroup sizes that may be of particular interest in fairness research and dataset auditing work. All values of `AVG-SEPFID` reported in the results are *out of sample*; i.e. the `AVG-SEPFID` values are computed on a test set that was not used to optimize the subgroups. Datasets were split into $80 - 20$ train-test split except for Student which was split $50 - 50$ due to its small size. Across all datasets, when the `FID` was LIME or SHAP, we set $h$ to be a random forest, when it was GRAD we used logistic regression as it requires a classifier whose outputs are differentiable in the inputs. The exact choice of classifier does not have any notable impact on the outcomes as discussed in Appendix K. Due to computation constraints, GRAD was only tested on the COMPAS R dataset. We defer the details in how we implemented the importance notions and Algorithm 1 to Appendix G.

**Linear Feature Importance Disparity**: In addition to the 3 separable notions of `FID`, we also studied an approach for a non-separable notion of importance. Linear regression (labeled LR in results) is a popular model that is inherently interpretable; the coefficients of a weighted least squares (WLS) solution represent the importance of each feature. We can thus define another variant of `FID`, the linear feature importance disparity (`LIN-FID`), as the difference in the WLS coefficient of feature $f_j$ on subgroup $g$ and on the dataset $X^n$. As `LIN-FID` is differentiable with respect to $g$, we are able to find a locally optimal $g$ with high `LIN-FID` using a non-convex optimizer; we used ADAM. For details and proofs, see Appendix F.

### 4.2 EXPERIMENTAL RESULTS

Table 1 summarizes the results of the experiments, which are visualized in Figure 2 on a log-ratio scale for better cross-notion comparison. Across each dataset and importance notion, our methods were able to find subgroups with high `FID`, often differing by orders of magnitude. For example, on Folktables with LIME as the importance notion, there is a subgroup on which `age` is on average 225 times more important than it is for the whole population. Table 1 also provides the defining features, listed as the sensitive features which have the largest coefficients in $g$.

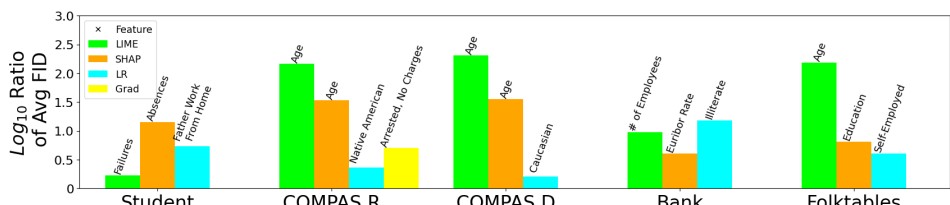

Figure 2: Summary of the highest `FID`s found for each (dataset, method). This is displayed as $\left| log_{10}(R) \right|$ where $R$ is the ratio of average importance per data point in $g^*$ to the average importance on $X$ for separable notions, or the ratio of coefficients for `LIN-FID`. This scale allows comparison across different importance notions. The feature associated with each $g^*$ is written above the bar.

A natural follow up question that arises from this experiment is what does the distribution of `FID`s look like for a given dataset? Figure 3 shows a distribution of the 10 features on the Bank dataset with the highest `FID` values. As we can see, there are a few features where large `FID` subgroups can be found, but it tails off significantly. This pattern is replicated across all datasets and feature importance notions. This is a positive result for practical uses, as an analyst or domain expert can focus on a handful of features that perform drastically differently when auditing a dataset for fairness concerns.

Table 1: Summary of the subgroup with highest `AVG-SEPFID` for each experiment along with the corresponding feature, subgroup size, and defining features. Experiments were run across multiple $(\alpha_L, \alpha_U)$ ranges with the highest `AVG-SEPFID` found being displayed. $\mu(F)$ is the average feature importance value on the specified group.

| Dataset | Notion | Feature $f_j$ | $\mu(F(f_j, X))$ | $\mu(F(f_j, g))$ | $|g|$ | Defining Features |
|---|---|---|---|---|---|---|
| Student | LIME | Failures | $-.006$ | $-.011$ | .01 | Alcohol Use, Urban Home |
| | SHAP | Absences | $-.15$ | $-2.1$ | .02 | Parent Status, Urban Home |
| | LR | Father WFH | 21.7 | $-4.0$ | .03 | Alcohol Use, Health |
| COMPAS R | LIME | Age | .0009 | $-.14$ | .05 | Native-American |
| | SHAP | Age | .012 | .41 | .04 | Asian-American |
| | LR | Native American | .5 | 1.17 | .04 | Asian/Hispanic-American |
| | GRAD | Arrest, No Charge | .07 | .16 | .05 | Asian/Native-American |
| COMPAS D | LIME | Age | $-.0003$ | $-.06$ | .02 | Native/Black-American |
| | SHAP | Age | .06 | 2.35 | .07 | Black/Asian-American |
| | LR | Caucasian | 6.7 | 10.7 | .04 | Native-American |
| Bank | LIME | # of Employees | $-.003$ | .03 | .03 | Marital Status |
| | SHAP | Euribor Rate | $-.004$ | .016 | .03 | Marital Status |
| | LR | Illiterate | $-.07$ | $-.0045$ | .01 | Age, Marital Status |
| Folktables | LIME | Age | $-.0007$ | $-.11$ | .21 | Marital Status |
| | SHAP | Education | .023 | .15 | .03 | Asian-American |
| | LR | Self-Employed | $-.26$ | $-.06$ | .02 | White-American |

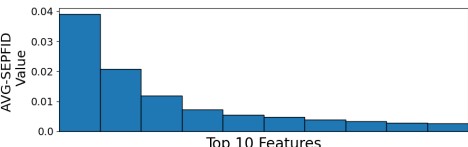

Figure 3: Distribution of `AVG-SEPFID` on the top features from the BANK dataset using LIME. We see a sharp drop off in `AVG-SEPFID`. This pattern is replicated in all datasets and notions.

In Figure 4, we highlight a (feature, subgroup, method) pair on the COMPAS R dataset representative of the kinds of bias this work could expose. In our classification model explained using GRAD, the feature `arrested-but-with-no-charges` is highly important when predicting `two-year-recidivism` on $X$. However, it carries almost no importance on $g^*$ which is largely defined by Native-American males. In every dataset and feature importance notion, we found similar examples exposing some form of potential bias, see Appendix H for more examples. These examples, in conjunction with the results reported in Table 1, highlight the usefulness of our method in finding subgroups where a concerned analyst or domain expert could dig deeper to determine how biases might be manifesting themselves in the data and how to correct for them.

### 4.3 FAIRNESS METRICS

While large `AVG-SEPFID` values with respect to a given feature and importance notion do not imply disparities in common fairness metrics, which are not typically defined in terms of a specific reference feature, it is natural to ask if these notions are correlated: do subgroups with large `AVG-SEPFID` have large disparities in fairness metrics, and do subgroups that have large disparities in fairness metrics have particularly large `AVG-SEPFID` values for some feature?

We examine the first question in Table 2. We find that these high `AVG-SEPFID` subgroups tend to have significant disparities in these traditional fairness metrics. Although the metrics are not always *worse* on $g$, this reinforces the intuition that subgroups with high `AVG-SEPFID` require greater scrutiny. In Table 3 we study the reverse question, where we used the GerryFair code of Kearns et al. (2018) to find rich subgroups that maximally violate FPR disparity, and then compute the `AVG-SEPFID` on those subgroups. We find that they also have features with high `AVG-SEPFID`, albeit not as large as those found by Algorithm 1, which explicitly optimizes for `AVG-SEPFID`. These two results highlight the usefulness of our method in identifying potentially high risk subgroups.

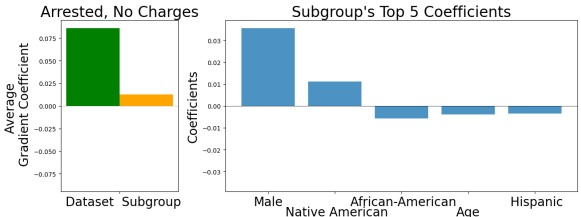

Figure 4: Exploring a key subgroup/feature pair for COMPAS R using GRAD. The first graph compares $F(f_j, X)$ and $F(f_j, g^*)$. The second graph shows the 5 largest coefficients of $g^*$.

Table 2: Observing fairness metrics of high `AVG-SEPFID` subgroups. COMPAS D and Student were excluded since they use non-binary $y$, making classification metrics less comparable. We measured the three fairness types outlined by Barocas et al. (2019): $P(\hat{Y} = 1)$, true/false positive rates, and expected calibration error. Each *<metric>*$_\Delta$ corresponds to the metric on $g$ minus the metric on $X$.

| Dataset | Notion | $F$ | Defining Features of $g$ | $\hat{Y}_\Delta$ | $TPR_\Delta$ | $FPR_\Delta$ | $ECE_\Delta$ |
|---|---|---|---|---|---|---|---|
| COMPAS R | LIME | Age | Native-American | $-.16$ | $-.18$ | .02 | .24 |
| | SHAP | Age | Asian-American | .37 | .35 | .02 | $-.12$ |
| | GRAD | Arrest, No Charge | Asian/Native-American | $-.19$ | $-.22$ | .03 | .24 |
| Bank | LIME | # of Employees | Marital Status | .11 | .06 | .05 | $-.15$ |
| | SHAP | Euribor Rate | Marital Status | .11 | .06 | .05 | $-.17$ |
| Folktables | LIME | Age | Marital Status | $-.15$ | $-.11$ | $-.04$ | .19 |
| | SHAP | Education | Asian-American | .17 | .16 | .01 | $-.09$ |

Table 3: Comparing top features and respective `AVG-SEPFID` of $g$ found via our method (`AVG-SEPFID`$_{FID}$) and found by Kearns et al. (2018) (`AVG-SEPFID`$_{gerry}$). As in Table 2, COMPAS D and Student were excluded.

| Dataset | Notion | $F_{FID}$ | `AVG-SEPFID`$_{FID}$ | $F_{gerry}$ | `AVG-SEPFID`$_{gerry}$ |
|---|---|---|---|---|---|
| COMPAS R | LIME | Age | .14 | Age | .04 |
| | SHAP | Age | .4 | Age | .06 |
| | GRAD | Arrest, No Charge | .09 | Male | .03 |
| Bank | LIME | # of Employees | .03 | # of Employees | .008 |
| | SHAP | Euribor Rate | .016 | Emp Var Rate | .004 |
| Folktables | LIME | Age | .11 | Age | .05 |
| | SHAP | Education | .13 | Age | .05 |

## 5 LIMITATIONS

Importantly, we eschew any broader claims that large `FID` *necessarily* implies a mathematical conclusion about the *fairness* of the underlying classification model in all cases. It is known that even the most popular and natural fairness metrics are impossible to satisfy simultaneously, and so we would run up against the problem of determining what it means for a model to be *fair* Chouldechova (2017); Kleinberg et al. (2017). By detecting anomalous subgroups with respect to feature importance, our approach can signal to a domain expert that perhaps there are issues such as feature collection or measurement bias. This will facilitate the next steps of testing the resulting hypotheses, and ultimately intervening to address disparities and improve fairness outcomes. Concerns about the stability and robustness of the most widely used feature importance notions, including the ones we study, have been raised Dai et al. (2022); Agarwal et al. (2022a); Alvarez-Melis & Jaakkola (2018); Bansal et al. (2020); Dimanov et al. (2020); Slack et al. (2020) and these notions are often at odds with each other, so none can be considered definitive Krishna et al. (2022). Regardless of these limitations, these notions are used widely in practice today, and are still useful as a diagnostic tool as we propose here in order to uncover potentially interesting biases. Lastly, our methods, like nearly all prior works on fairness, require tabular datasets that have defined the sensitive features apriori, a process more difficult in text or image datasets where bias is still a concern Buolamwini & Gebru (2018); Bolukbasi et al. (2016). Overall, the methods developed here represent a part of the algorithmic toolkit that domain experts may use in rooting out bias.

## 6 REPRODUCIBILITY

Specific details for the experiments such as the hyperparameters used are available in Appendix G. The source code used for these experiments is provided in the supplementary material. Specifically, `run_separable.py` and `run_linear.py` are the scripts where the importance notion (Appendix G.2), dataset (Appendix G.3), and other parameters are specified before running. The `experiments/` directory contains scripts used for the comparison of rich and marginal subgroups as seen in Appendix I and for the fairness comparison experiments in Subsection 4.3.

## 7 ETHICAL REVIEW

There were no substantial ethical issues that came up during this research process. The datasets used are all publicly available, de-identified, and have frequently been used in fair machine learning research. There was no component of this research that sought to re-identify the data or use it in any fashion other than to test our methodology.

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

# A    ADDITIONAL RELATED WORK

**Fairness in Machine Learning**. Much of the work in fairness in machine learning typically concerns the implementation of a new fairness notion in a given learning setting; either an individual fairness notion Dwork et al. (2012); Joseph et al. (2018), one based on equalizing a statistical rate across protected subgroups Hardt et al. (2016); Pleiss et al. (2017), or one based on an underlying causal model Kusner et al. (2017). With a given notion of fairness in hand, approaches to learning fair classifiers can be typically classified as "in-processing", or trying to simultaneously learn a classifier and satisfy a fairness constraint, "post-processing" which takes a learned classifier and post-processes it to satisfy a fairness definition Hardt et al. (2016), or most closely related to the motivation behind this paper, pre-processing the data to remove bias. Existing work on dataset bias serve as high level motivation for our work.

**Feature Importance Notions**. The local explanation methods mentioned in Section 1.2 include model-agnostic methods like LIME or SHAP Ribeiro et al. (2016); Lundberg & Lee (2017), methods like saliency maps Simonyan et al. (2013); Sundararajan et al. (2017); Baehrens et al. (2010) that require $h$ to be differentiable in $x$, or model-specific methods that depend on the classifier. In addition to these explanation methods, there are also global methods that attempt to explain the entire model behavior and so can be run on the entire subgroup. Our `LIN-FID` method as described in Appendix F is a global method that relies on training an inherently interpretable model (linear regression) on the subgroup and inspecting its coefficients. Other inherently interpretable models that could be used to define a notion of subgroup importance include decision trees Quinlan (1986) and generalized additive models Liu et al. (2022).

**Fairness and Interpretability**. Although no existing work examines the role of feature importance notions in detecting disparities in rich subgroups, there is a small amount of existing work examining explainability in the context of fairness. The recent Grabowicz et al. (2022) formalizes induced discrimination as a function of the SHAP values assigned to sensitive features, and proposes a method to learn classifiers where the protected attributes have low influence. Begley et al. (2020) applies a similar approach, attributing a models overall unfairness to its individual features using the Shapley value, and proposing an intervention to improve fairness. Ingram et al. (2022) examines machine learning models to predict recidivism, and empirically shows tradeoffs between model accuracy, fairness, and interpretability.

Additionally, Lundberg (2020) decomposes feature attribution explanations and fairness metrics into additive components and observes the relationship between the fairness metrics and input features. Our work does not try to decompose fairness metrics into additive components and also focuses on non-additive feature explanations. Furthermore, our consideration of rich subgroups is a novel addition to the space.

# B    PROOF OF THEOREM 1

We start by showing that for the unconstrained problem, computing the subgroup $g_j^*$ that maximizes $\texttt{FID}(f_j, g, h)$ over $\mathcal{G}$ can be computed in two calls to $\text{CSC}_\mathcal{G}$ when $F$ is separable.

**Lemma 1.** *If $F$ is separable and $CSC_\mathcal{G}$ is a CSC oracle for $\mathcal{G}$, then for any feature $f_j$, $g_j^*$ can be computed with two oracle calls.*

*Proof.* By definition $g_j^* = \text{argmax}_{g \in \mathcal{G}}\texttt{FID}(j, g) = \text{argmax}_{g \in \mathcal{G}}|F(f_j, X^n, h) - F(f_j, g, h)| = \text{argmax}_{g \in \{g^+, g^-\}}\texttt{FID}(j, g)$, where $g^+ = \text{argmax}_{g \in \mathcal{G}}F(f_j, g, h), g^- = \text{argmin}_{g \in \mathcal{G}}F(f_j, g, h)$. By the definition of separability, we can write

$$F(f_j, g(X^n), h) = \sum_{X \in g(X^n)} F'(f_j, X, h) = \sum_{i=1}^{n} g(X_i)F'(f_j, X_i, h)$$

Then letting $c_k^0 = 0$ and $c_k^1 = -F'(f_j, X_k, h)$ for $k = 1, \ldots n$, we see that $g^+ = \text{CSC}_g((c_k^0, c_k^1)), g^- = \text{CSC}_g((c_k^0, -c_k^1))$. This establishes the claim. $\square$

Theorem 1: Let $F$ be a separable notion, fix a classifier $h$, subgroup class $\mathcal{G}$, and oracle $\text{CSC}_\mathcal{G}$. Then fixing a feature of interest $f_j$, we will run Algorithm 1 twice; once with $\texttt{FID}$ given by $F$, and

once with `FID` given by $-F$. Let $\hat{p}_{\mathcal{G}}^T$ be the distribution returned after $T = O(\frac{4n^2B^2}{\nu^2})$ iterations by Algorithm 1 that achieves the larger value of $\mathbb{E}[\texttt{FID}(j, g)]$. Then:

$$\texttt{FID}(j, g_j^*) - \mathbb{E}_{g \sim \hat{p}_{\mathcal{G}}^T}[\texttt{FID}(j, g)] \leq \nu$$

$$|\Phi_L(g)|, |\Phi_U(g)| \leq \frac{1 + 2\nu}{B} \tag{3}$$

*Proof.* We start by transforming our constrained optimization into optimizing a $\min - \max$ objective. The $\min$ player, referred to as the *subgroup player* will be solving a CSC problem over the class $\mathcal{G}$ at each iteration, while the $\max$ player, called the *dual player*, will be adjusting the dual weights $\lambda$ on the two constraints using the exponentiated gradient algorithm Kivinen & Warmuth (1997). By Lemma 2 Freund & Schapire (1996), we know that if each player implements a *no-regret* strategy, then the error of subgroup found after $T$ rounds is sub-optimal by at most the average cumulative regret of both players. The regret bound for the exponentiated gradient descent ensures this occurs in *poly(n)* rounds.

As in Kearns et al. (2018); Agarwal et al. (2018), we first relax Equation 1 to optimize over all *distributions* over subgroups, and we enforce that our constraints hold in expectation over this distribution. Our new optimization problem becomes:

$$\min_{p_g \in \Delta(\mathcal{G})} \quad \mathbb{E}_{g \sim p_g}[\sum_{i=1}^{n} g(x_i)F'(f_j, x_i, h)]$$

$$\text{s.t.} \quad \mathbb{E}_{g \sim p_g}[\Phi_L(g)] \leq 0 \tag{4}$$

$$\mathbb{E}_{g \sim p_g}[\Phi_U(g)] \leq 0$$

We note that while $|\mathcal{G}|$ may be infinite, the number of distinct labelings of $X$ by elements of $\mathcal{G}$ is finite; we denote the number of these by $|\mathcal{G}(X)|$. Then since Equation 4 is a finite linear program in $|\mathcal{G}(X)|$ variables, it satisfies strong duality, and we can write:

$$(p_g^*, \lambda^*) = \text{argmin}_{p_g \in \Delta(\mathcal{G})} \text{argmax}_{\lambda \in \Lambda} \mathbb{E}_{g \sim p_g}[L(g, \lambda)] = \text{argmin}_{p_g \in \Delta(\mathcal{G})} \text{argmax}_{\lambda \in \Lambda} L(p_g, \lambda)$$

$$\text{with} \quad L(g, \lambda) = \sum_{x \in X} g(x)F(f_j, x, h) + \lambda_L \Phi_L + \lambda_U \Phi_U, \quad L(p_g, \lambda) = \mathbb{E}_{g \sim p_g}[L(g, \lambda)]$$

As in Kearns et al. (2018) $\Lambda = \{\lambda \in \mathbb{R}^2 \mid \|\lambda\|_1 \leq B\}$ is chosen to make the domain compact, and does not change the optimal parameters as long as $B$ is sufficiently large, i.e. $\|\lambda^*\|_1 \leq B$. In practice, this is a hyperparameter of Algorithm 1, similar to Agarwal et al. (2018); Kearns et al. (2018). Then we follow the development in Agarwal et al. (2018); Kearns et al. (2018) to show that we can compute $(p_g^*, \lambda^*)$ efficiently by implementing *no-regret* strategies for the subgroup player $(p_g)$ and the dual player $(\lambda)$.

Formally, since $\mathbb{E}_{g \sim p_g}[L(g, \Lambda)]$ is bi-linear in $p_g, \lambda$, and $\Lambda, \Delta(\mathcal{G})$ are convex and compact, by Sion's minimax theorem Kindler (2005):

$$\min_{p_g \in \Delta(G)} \max_{\lambda \in \Lambda} L(p_g, \lambda) = \max_{\lambda \in \Lambda} \min_{p_g \in \Delta(G)} L(p_g, \lambda) = \text{OPT} \tag{5}$$

Then by Theorem 4.5 in Kearns et al. (2018), we know that if $(p_g^*, \lambda^*)$ is a $\nu$-approximate min-max solution to Equation 5 in the sense that

$$\text{if:} \quad L(p_g^*, \lambda^*) \leq \min_{p \in \Delta(\mathcal{G})} L(p, \lambda^*) + \nu, L(p_g, \lambda) \geq \max_{\lambda \in \Lambda} L(p^*, \lambda),$$

$$\text{then:} \quad F(f_j, p_g^*, h) \leq OPT + 2\nu, \quad |\Phi_L(g)|, |\Phi_U(g)| \leq \frac{1 + 2\nu}{B} \tag{6}$$

So in order to compute an approximately optimal subgroup distribution $p_g^*$, it suffices to compute an approximate min-max solution of Equation 5. In order to do that we rely on the classic result of

Freund & Schapire (1996) that states that if the subgroup player best responds, and if the dual player achieves low regret, then as the average regret converges to zero, so does the sub-optimality of the average strategies found so far.

**Lemma 2** (Freund & Schapire (1996)). *Let $p_1^\lambda, \ldots p_T^\lambda$ be a sequence of distributions over $\Lambda$, played by the dual player, and let $g^1, \ldots g^T$ be the subgroup players best responses against these distributions respectively. Let $\hat{\lambda}_T = \frac{1}{T} \sum_{t=1}^T p_t^\lambda, \hat{p}_g = \frac{1}{T} \sum_{t=1}^T g_t$. Then if*

$$\sum_{t=1}^T \mathbb{E}_{\lambda \sim p_t^\lambda}[L(g_t, \lambda)] - \min_{\lambda \in \Lambda} \sum_{t=1}^T [L(g_t, \lambda)] \leq \nu T,$$

*Then $(\hat{\lambda}_T, \hat{p}_g)$ is a $\nu$-approximate minimax equilibrium of the game.*

To establish Theorem 1, we need to show (i) that we can efficiently implement the subgroup players best response using $CSC_\mathcal{G}$ and (ii) we need to translate the regret bound for the dual players best response into a statement about optimality, using Lemma 2. Establishing $(i)$ is immediate, since at each round $t$, if $\lambda_{t,0} = \mathbb{E}_{p_t^\lambda}[\lambda_L], \lambda_{t,1} = \mathbb{E}_{p_t^\lambda}[\lambda_U]$, then the best response problem is:

$$\mathrm{argmin}_{p_g \in \Delta(G)} \mathbb{E}_{g \sim p_g}[\sum_{x \in X} g(x) F(f_j, x, h) + \lambda_{t,0} \Phi_L + \lambda_{t,1} \Phi_U]$$

Which can further be simplified to:

$$\mathrm{argmin}_{g \in G} \sum_{x \in X} g(x)(F(f_j, x, h) - \lambda_L + \lambda_U) \tag{7}$$

This can be computed with a single call of $CSC_\mathcal{G}$, as desired. To establish (ii), the no-regret algorithm for the dual player's distributions, we note that at each round the dual player is playing online linear optimization over 2 dimensions. Algorithm 1 implements the exponentiated gradient algorithm Kivinen & Warmuth (1997), which has the following guarantee proven in Theorem 1 of Agarwal et al. (2018), which follows easily from the regret bound of exponentiated gradient Kivinen & Warmuth (1997), and Lemma 2:

**Lemma 3** (Agarwal et al. (2018)). *Setting $\eta = \frac{\nu}{2n^2 B}$, Algorithm 1 returns $\hat{p}_\lambda^T$ that is a $\nu$-approximate min-max point in at most $O(\frac{4n^2 B^2}{\nu^2})$ iterations.*

Combining this result with Equation 5 completes the proof.

$\square$

## C  PROOF OF AVG-SEPFID PRIMITIVE

In Section 3, we presented our approach that optimizing for FID constrained across a range of subgroup sizes will allow us to efficiently optimize for AVG-SEPFID. We provide a more complete proof of that claim here:

Let $g^*$ be the subgroup that maximizes AVG-SEPFID. Without loss of generality, $g^* = \mathrm{argmax}_{g \in \mathcal{G}} \frac{1}{n|g|} \sum g(x) F'(f_j, X, h)$ (we drop the absolute value because we can also set $F' = -F$). Then it is necessarily true, that $g^*$ also solves the constrained optimization problem $\mathrm{argmax}_{g \in \mathcal{G}} \frac{1}{n} \sum g(x) F'(f_j, X, h)$ such that $|g| = |g^*|$, where we have dropped the normalizing term $\frac{1}{|g|}$ in the objective function, and so we are maximizing the constrained FID.

Now consider an interval $I = [|g^*| - \alpha, |g^*| + \alpha]$, and suppose we solve $g_I^* = \mathrm{argmax}_{g \in \mathcal{G}} \frac{1}{n} \sum g(x) F'(f_j, X, h)$ such that $g \in I$. Then since $g^* \in I$, we know that $\frac{1}{n} \sum g^* F'(f_j, X, h) \leq \frac{1}{n} \sum g_I^*(x) F'(f_j, X, h)$. This implies that:

$$
\begin{aligned}
\texttt{AVG-SEPFID}(g_I^*) &\geq \frac{1}{|g_I^*|}\frac{1}{n}\sum g^*(x)F'(f_j, X, h) \\
&= \texttt{AVG-SEPFID}(g^*) + \left(\frac{1}{|g_I^*| + \alpha} - \frac{1}{|g_I^*|}\right)\texttt{FID}(g^*) \\
&= \texttt{AVG-SEPFID}(g^*) - \frac{\alpha}{|g^*|(|g^*| + \alpha)}\cdot\texttt{FID}(g^*)
\end{aligned}
$$

Given the above derivation, as $\alpha \to 0$, we have $\texttt{AVG-SEPFID}(g_I^*) \to \texttt{AVG-SEPFID}(g^*)$.

Hence we can compute a subgroup $g$ that approximately optimizes the $\texttt{AVG-SEPFID}$ if we find an appropriately small interval $I$ aroudn $|g^*|$. Since the discretization in Section 3 covers the unit interval, we are guaranteed for sufficiently large $n$ to find such an interval.

## D  COST SENSITIVE CLASSIFIER, $\text{CSC}_\mathcal{G}$

**Definition 4.** *(Cost Sensitive Classification) A Cost Sensitive Classification (CSC) problem for a hypothesis class $\mathcal{G}$ is given by a set of $n$ tuples $\{(X_i, c_i^0, c_i^1)\}_{i=1}^n$, where $c_i^0$ and $c_i^1$ are the costs of assigning labels $0$ and $1$ to $X_i$ respectively. A CSC oracle finds the classifier $\hat{g} \in \mathcal{G}$ that minimizes the total cost across all points:*

$$
\hat{g} = \operatorname*{argmin}_{g \in \mathcal{G}}\sum_i\left(g(X_i)c_i^1 + (1 - g(X_i))c_i^0\right) \tag{8}
$$

---

**Algorithm 2** $\text{CSC}_\mathcal{G}$

---

**Input:** Dataset $X \subset \mathbb{R}^{d_{sens}} \times \mathbb{R}^{d_{safe}}$, costs $(c^0, c^1) \in \mathbb{R}^n$
Let $X_{sens}$ consist of the sensitive attributes $x$ of each $(x, x') \in X$.
Train linear regressor $r_0 : \mathbb{R}^{d_{sens}} \to \mathbb{R}$ on dataset $(X_{sens}, c^0)$     ▷ learn to predict the cost $c^0$
Train linear regressor $r_1 : \mathbb{R}^{d_{sens}} \to \mathbb{R}$ on dataset $(X_{sens}, c^1)$     ▷ learn to predict the cost $c^1$
Define $g((x, x')) := \mathbf{1}\{(r_0 - r_1)(x) > 0\}$     ▷ predict 0 if the estimated $c_0 < c_1$
Return $g$

---

## E  NP-COMPLETENESS

We will show below that the fully general version of this problem (allowing any poly-time $F$) is NP complete. First, we will define a decision variant of the problem:

$$
\delta_{X,F,h,A} = \max_{g \in \mathcal{G}, f_j}\left(|F(f_j, g, h) - F(f_j, X, h)|\right) \geq A
$$

Note that a solution to the original problem trivially solves the decision variant. First, we will show the decision variant is in NP, then we will show it is NP hard via reduction to the max-cut problem.

**Lemma 4.** *The decision version of this problem is in NP.*

*Proof.* Our witness will be the subset $g$ and feature $f_j$ such that

$$
(|F(f_j, g, h) - F(f_j, X, h)|) \geq A
$$

Given these 2, evaluation of the absolute value is polytime given that $F$ is polytime, so the solution can be verified in polytime. $\qquad\square$

**Lemma 5.** *The decision version of this problem is NP hard.*

*Proof.* We will define our variables to reduce our problem to maxcut$(Q, k)$. Given a graph defined with $V$, $E$ as the vertex and edge sets of $Q$ (with edges defined as pairs of vertices), we will define our $F$, $X$, $G$, $A$, and $h$ as follows:

$$
\begin{aligned}
X &= V \\
h &= \text{constant classifier, maps every value to 1} \\
\mathcal{G} &= \mathcal{P}(V) \text{ i.e. all possible subsets of vertices} \\
F(f_j, g, h) &= |x \in E : x[0] \in g, x[1] \in g^c| \\
&\quad \text{--i.e. } F(j, g, h) \text{ returns the number of} \\
&\quad \text{edges cut by a particular subset, ignoring} \\
&\quad \text{its first and third argument.} \\
&\quad \text{(this is trivially computable in polynomial} \\
&\quad \text{time by iterating over the set of edges).} \\
A &= k
\end{aligned}
$$

Note that $F(f_j, X, h) = 0$ by definition, and that $F \geq 0$. Therefore, $|F(f_j, g, h) - F(f_j, X, h)| = F(f_j, g, h)$, and we see that $(|F(f_j, g, h) - F(f_j, X, h)|) \geq A$ if and only if $g$ is a subset on $Q$ that cuts at least $A = k$ edges. Therefore an algorithm solving the decision variant of the feature importance problem also solves maxcut. $\square$

## F   LINEAR FEATURE IMPORTANCE DISPARITY

The *non-separable* FID notion considered in this paper corresponds to training a model that is inherently interpretable on only the data in the subgroup $g$, and comparing the influence of feature $j$ to the influence when trained on the dataset as a whole. Since all of the points in the subgroup can interact to produce the interpretable model, this notions typically are not separable. Below we formalize this in the case of linear regression, which is the non-separable notion we investigate in the experiments.

**Definition 5.** *(Linear Feature Importance Disparity).   Given a subgroup $g$, let $\theta_g = \inf_{\theta \in \mathbb{R}^d} \mathbb{E}_{(X,y) \sim \mathcal{R}}[g(X)(\theta'X - y)^2]$, and $\theta_\mathcal{R} = \inf_{\theta \in \mathbb{R}^d} \mathbb{E}_{(X,y) \sim \mathcal{R}}[(\theta'X - y)^2]$. Then if $e_j$ is the $j^{th}$ basis vector in $\mathbb{R}^d$, we define the* linear feature importance disparity *(LIN-FID) by*

$$
\text{LIN-FID}(j, g) = |(\theta_g - \theta_\mathcal{R}) \cdot e_j|
$$

LIN-FID$(j, g)$ is defined as the difference between the coefficient for feature $j$ when training the model on the subgroup $g$, versus training the model on points from $\mathcal{R}$. Expanding Definition 5 using the standard weighted least squares estimator (WLS), the feature importance for a given feature $f_j$ and subgroup $g(X)$ is:

$$
F_{lin}(j, g) = \left((Xg(X)X^T)^{-1}(X^T g(X)Y)\right) \cdot e_j, \tag{9}
$$

Where $g(X)$ is a diagonal matrix of the output of the subgroup function. The coefficients of the linear regression model on the dataset $X$ can be computed using the results from ordinary least squares (OLS): $(XX^T)^{-1}(X^TY) \cdot e_j$.

We compute $\text{argmax}_{g \in G}\text{LIN-FID} = \text{argmax}_{g \in G}|F_{lin}(j, X^n) - F_{lin}(j, g)|$ by finding the minimum and maximum values of $F_{lin}(j, g)$ and choosing the one with the larger difference. For the experiments in Section 4, we use logistic regression as the hypothesis class for $g$ because it is non-linear enough to capture complex relationships in the data, but maintains interpretability in the form of its coefficients, and importantly because Equation 9 is then differentiable in the parameters $\theta$ of $g(X) = \sigma(X \cdot \theta), \sigma(x) = \frac{1}{1+e^{-x}}$. Since Equation 9 is differentiable in $\theta$, we can use non-convex optimizers like SGD or ADAM to maximize Equation 9 over $\theta$.

While this is an appealing notion due to its simplicity, it is not relevant unless the matrix $Xg(X)X^T$ is of full rank. We ensure this first by lower bounding the size of $g$ via a size penalty term $P_{size} = \max(\alpha_L - |g(X_{train})|, 0) + \max(|g(X_{train})| - \alpha_U, 0)$, which allows us to provide $\alpha$ constraints

in the same manner as in the separable approach. We also add a small $l_2$ regularization term $\epsilon I$ to $X^T g(X) X$. This forces the matrix to be invertible, avoiding issues with extremely small subgroups. Incorporating these regularization terms, Equation 9 becomes:

$$F_{lin}(j, g) = \lambda_s \cdot \left( (X\sigma(X \cdot \theta_L^T) X^T + \epsilon I)^{-1} (X^T \sigma(X \cdot \theta_L^T) Y) \cdot e_j \right) + \lambda_c \cdot P_{size} \qquad (10)$$

We note that `LIN-FID` is a similar notion to that of LIME Ribeiro et al. (2016), but LIME estimates a local effect around each point which is then summed to get the effect in the subgroup, and so it is *separable*. It is also the case that $F_{lin}$ is non-convex as shown below:

**Lemma 6.** $F_{lin}$ *as defined in Equation 9 is non-convex.*

*Proof.* We will prove this by contradiction. Assume $F_{lin}$ is convex, which means the Hessian is positive semi-definite everywhere. First we will fix $(Xg(X)X^T))^{-1}$ to be the identity matrix, which we can do without loss of generality by scaling $g$ by a constant. This scaling will not affect the convexity of $F_{lin}$.

Now, we have the simpler form of $F_{lin} = (X^T g(X) Y) \cdot e_j$. We then can compute the values of the Hessian:

$$\frac{\partial F^2}{\partial^2 g} = (X^T g''(X) Y) \cdot e_j$$

Consider the case where $X^T$ is a $2 \times 2$ matrix with rows $1, 0$ and $0, -1$ and $Y$ is a vector of ones. If $g$ weights the second column (i.e. feature) greater than the first, then the output Hessian will be positive semi-definite. But if $g$ weights the first column greater than the first, then it will be negative semi-definite. Since the Hessian is not positive semi-definite everywhere, $F_{lin}$ must be non-convex over the space of $g$. $\qquad \square$

This means the stationary point we converge to via gradient descent may only be locally optimal. In Section 4, we optimize Equation 10 using the ADAM optimizer Kingma & Ba (2015). Additional details about implementation and parameter selection are in Appendix G. Despite only locally optimal guarantees, we were still able to find (feature, subgroup) pairs with high `LIN-FID` for all datasets.

## G    EXPERIMENTAL DETAILS

### G.1    ALGORITHMIC DETAILS

**Separable Case.** In order to implement Algorithm 1 over a range of $[\alpha_L, \alpha_U]$ values, we need to specify our dual norm $B$, learning rate $\eta$, number of iterations used $T$, rich subgroup class $\mathcal{G}$, and the associated oracle $\text{CSC}_{\mathcal{G}}$. We note that for each feature $f_j$, Algorithm 1 is run twice; one corresponding to maximizing $\text{FID}(f_j, g, h)$ and the other minimizing it. Note that in both cases our problem is a minimization, but when maximizing we simply negate all of the point wise feature importance values $F(f_j, x_i, h) \to -F(f_j, x_i, h)$. In all experiments our subgroup class $\mathcal{G}$ consists of linear threshold functions over the sensitive features: $\mathcal{G} = \{\theta \in \mathbb{R}^{d_{sens}} : \theta((x, x')) = \mathbf{1}\{\theta' x > 0\}$. We implement $\text{CSC}_{\mathcal{G}}$ as in Agarwal et al. (2018); Kearns et al. (2018) via linear regression, see Algorithm 2 in Appendix D. To ensure the dual player's response is strong enough to enforce desired size constraints, we empirically found that setting the hyperparameter $B = 10^4 \cdot \mu(f_j)$ worked well on all datasets, where $\mu(f_j)$ is the average absolute importance value for feature $j$ over $X$. We set the learning rate for exponentiated gradient descent to $\eta = 10^{-5}$. Empirical testing showed that $\eta \cdot B$ should be on the order of $\mu(f_j)$ or smaller to ensure proper convergence. We found that setting the error tolerance hyperparameter $\nu = .05 \cdot \mu(f_j) \cdot n \cdot \alpha_L$ worked well in ensuring good results with decent convergence time across all datasets and values of $\alpha$. For all datasets and methods we ran for at most $T = 5000$ iterations, which we observe empirically was large enough for `FID` values to stabilize and for $\frac{1}{T} \sum_{t=1}^{T} |g_t| \in [\alpha_L, \alpha_U]$, with the method typically converging in $T = 3000$ iterations or less. See Appendix M for a sample of convergence plots.

**Non-Separable Case.** For the non-separable approach, datasets were once again split into train and test sets. For Student, it was split 50-50, while COMPAS, Bank, and Folktables were split 80-20

train/test. The 50-50 split for Student was chosen so that a linear regression model would be properly fit on a small $g(X_{test})$. The parameter vector $\theta$ for a logistic regression classifier was randomly initialized with a PyTorch random seed of 0 for reproducability. We used an ADAM Kingma & Ba (2015) optimizer with a learning rate of .05 as our heuristic solver for the loss function.

To enforce subgroup size constraints, $\lambda_s P_{size}$ must be on a significantly larger order than $\lambda_c F_{lin}(j, g)$. Empirical testing found that values of $\lambda_s = 10^5$ and $\lambda_c = 10^{-1}$ returned appropriate subgroup sizes and also ensured smooth convergence. The optimizer ran until it converged upon a minimized linear regression coefficient, subject to the size constraints. Experimentally, this took at most 1000 iterations, see Appendix N for a sample of convergence plots. After solving twice for the minimum and maximum $F_{lin}(j, g)$ values and our subgroup function $g$ is chosen, we fit the linear regression on both $X_{test}$ and $g(X_{test})$ to get the final `FID`.

### G.2 `FID` NOTIONS

**LIME**: A random forest model $h$ was trained on dataset $X^n$. Then each data point along with the corresponding probability outputs from the classifier were input into the LIME Tabular Explainer Python module. This returned the corresponding LIME explanation values.

**SHAP**: This was done with the same method as LIME, except using the SHAP Explainer Python module.

**Vanilla Gradient**: Labeled as *GRAD* in charts, the vanilla gradient importance notion was computed using the Gradient method from the OpenXAI library Agarwal et al. (2022b). This notion only works on differentiable classifiers so in this case, $h$ is a logistic regression classifier. We found there was no substantial difference between the choice of random forest or logistic regression for $h$ when tested on other importance notions (see Section J). Due to constraints on computation time, this method was only tested on the COMPAS dataset (using `Two Year Recidivism` as the target variable).

**Linear Regression**: For the linear regression notion, the subgroup $g$ was chosen to be in the logistic regression hypothesis class. For a given subgroup $g(X)$, the weighted least squares (WLS) solution is found whose linear coefficients $\theta_g$ then define the feature importance value $e_j \cdot \theta_g$.

For details on the consistency of these importance notions, see Appendix O.

### G.3 DATASETS

These four datasets were selected on the basis of three criterion: (i) they all use features which could be considered *sensitive* to make predictions about individuals in a context where bias in a significant concern (ii) they are heavily used datasets in research on interpretability and fairness, and as such issues of bias in the datasets should be of importance to the community, and (iii) they trace out a range of number of datapoints and number of features and sensitive features, which we summarise in Table 4. For each dataset, we specified features that were "sensitive." That is, when searching for subgroups with high `FID`, we only considered rich subgroups defined by features generally covered by equal protection or privacy laws (e.g. race, gender, age, health data).

**Student**: This dataset aims to predict student performance in a Portugese grade school using demographic and familial data. For the purposes of this experiment, the target variable was math grades at the end of the academic year. Student was by far the smallest of the four datasets with 395 data points. The sensitive features in Student are `gender`, `parental status`, `address` (urban or rural), `daily alcohol consumption`, `weekly alcohol consumption`, and `health`. Age typically would be considered sensitive but since in the context of school, age is primarily an indicator of class year, this was not included as a sensitive feature. The categorical features `address`, `Mother's Job`, `Father's Job`, and `Legal Guardian` were one hot encoded.

**COMPAS**: This dataset uses a pre-trial defendant's personal background and past criminal record to predict risk of committing new crimes. To improve generalizability, we removed any criminal charge features that appeared fewer than 10 times. Binary counting features (e.g. `25-45 yrs old` or `5+ misdemeanors`) were dropped in favor of using the continuous feature equivalents. Additionally, the categorical variable `Race` was one-hot encoded. This brought the total number of features to 95. The sensitive features in COMPAS are `age`, `gender`, and `race` (Caucasian, African-American,

Asian, Hispanic, Native American, and Other). For COMPAS, we ran all methodologies twice, once using the binary variable, `Two Year Recidivism`, as the target variable and once using the continuous variable `Decile Score`. `Two Year Recidivism` is what the model is intended to predict and is labeled as *COMPAS R* in the results. Meanwhile, `Decile Score` is what the COMPAS system uses in practice to make recommendations to judges and is labeled as *COMPAS D* in the results.

**Bank**: This dataset looks at whether a potential client signed up for a bank account after being contacted by marketing personnel. The sensitive features in Bank are `age` and `marital status` (married, single, or divorced). The `age` feature in Bank is a binary variable representing whether the individual is above the age of 25.

**Folktables**: This dataset is derived from US Census Data. Folktables covers a variety of tasks, but we used the ACSIncome task, which predicts whether an individual makes more than $50k per year. The ACSIncome task is meant to mirror the popular Adult dataset, but with modifications to address sampling issues. For this paper, we used data from the state of Michigan in 2018. To reduce sparseness of the dataset, the `Place of Birth` feature was dropped and the `Occupation` features were consolidated into categories of work as specified in the official Census dictionary Bureau (2020), (e.g. people who work for the US Army, Air Force, Navy, etc. were all consolidated into `Occupation=Military`). The sensitive features in Folktables are `age`, `sex`, `marital status` (married, widowed, divorced, separated, never married/under 15 yrs old), and `race` (Caucasian, African-American, Asian, Native Hawaiian, Native American singular tribe, Native American general, Other, and 2+ races).

Table 4: Summary of Datasets

| Dataset | Data Points | # of Features | # of Sensitive Features |
|---|---|---|---|
| Student | 395 | 32 | 6 |
| COMPAS | 6172 | 95 | 8 |
| Bank | 30488 | 57 | 4 |
| Folktables Income | 50008 | 52 | 16 |

# H   MORE DISCUSSION OF HIGH FID SUBGROUPS

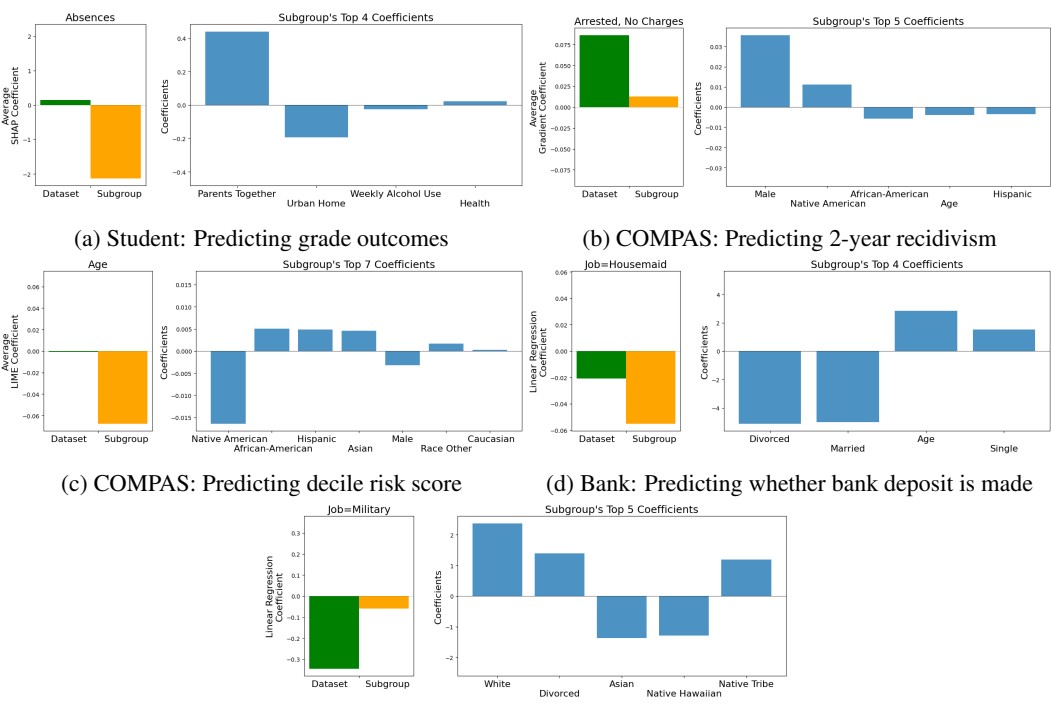

(a) Student: Predicting grade outcomes

(b) COMPAS: Predicting 2-year recidivism

(c) COMPAS: Predicting decile risk score

(d) Bank: Predicting whether bank deposit is made

(e) Folktables: Predicting income >$50k

Figure 5: Exploration of key subgroup/feature pairs found for each dataset. The first graph shows the change in feature importance from whole dataset to subgroup. The second graph shows the main coefficients that define the subgroup.

In Figure 5, we highlight selections of an interesting (feature, subgroup, method) pair for each dataset. Figure 5a shows that on the Student dataset the feature `absences` which is of near zero importance on the dataset as a whole, is very negatively correlated with student performance on a subgroup whose top 2 features indicate whether a student's parents are together, and if they live in an rural neighborhood. Figure 5b shows that on the COMPAS dataset with method GRAD, the feature `arrested-but-with-no-charges` is typically highly important when predicting `two-year-recidivism`. However, it carries significantly less importance on a subgroup that is largely defined as Native American males. When predicting the decile risk score on COMPAS, LIME indicates that age is not important on the dataset as a whole; however, for non-Native American, female minorities, older age can be used to explain a lower `Decile Score`. On the Bank dataset using `LIN-FID`, we see that a linear regression trained on points from a subgroup defined by older, single individuals, puts more importance on `job=housemaid` when predicting likelihood in signing up for an account. Finally on Folktables, we see that `LIN-FID` assigns much lower weight to the `job=military` feature among a subgroup that is mainly white and divorced people than in the overall dataset when predicting income. These interesting examples, in conjunction with the results reported in Table 1, highlight the usefulness of our method in finding subgroups where a concerned analyst/domain expert could dig deeper to determine how biases might be manifesting themselves in the data and if/how to correct for them.

# I   COMPARISON OF FID VALUES ON RICH VS. MARGINAL SUBGROUPS

To better justify the use of rich subgroups, we performed the same analysis but only searching over the marginal subgroup space. For each dataset and importance notion pair, we established the finite list of marginal subgroups defined by a single sensitive characteristic and computed the feature importance values on each of these subgroups. In Figure 6, we compare the maximal `AVG-SEPFID`

rich subgroups shown in Figure 2 to the maximal `AVG-SEPFID` marginal subgroup for the same feature. In about half of the cases, the `AVG-SEPFID` of the marginal subgroup was similar to the rich subgroup. In the other cases, expanding our subgroup classes to include rich subgroups defined by linear functions of the sensitive attributes enabled us to find a subgroup that had a higher `AVG-SEPFID`. For example, in Figure 6b, we can see that on the COMPAS R dataset using GRAD as the importance notion, `Arrested, No Charges` had a rich subgroup with `AVG-SEPFID` that was 4 times less than on the full dataset. However, we were unable to find any subgroup in the marginal space where the importance of the feature was nearly as different. In some cases in Figure 6, the marginal subgroup performs slightly better than the rich subgroup. This happens when using rich subgroups does not offer any substantial advantage over marginal subgroups, and the empirical error tolerance in Algorithm 1 stopped the convergence early.

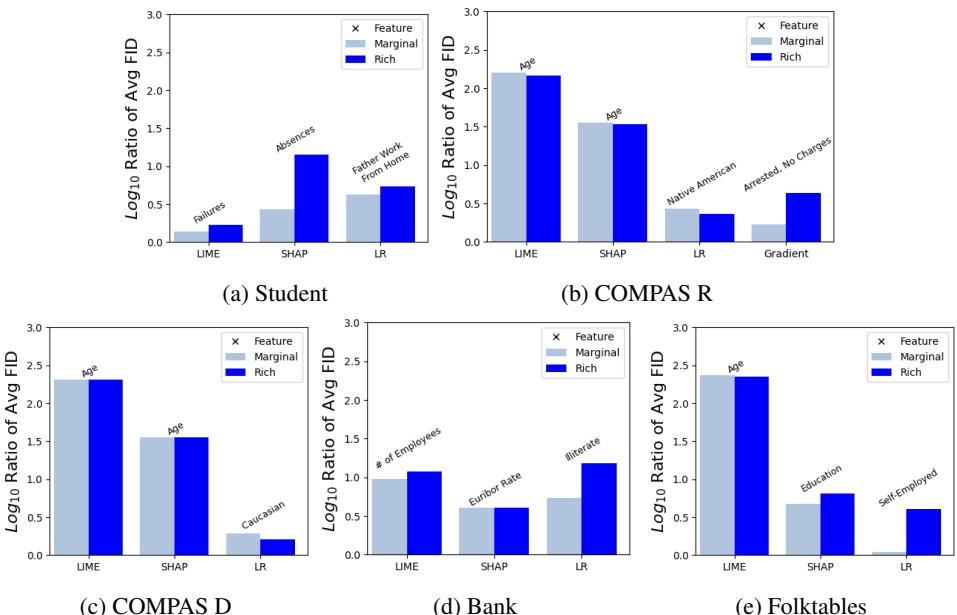

Figure 6: Comparison of the maximal `FID` rich subgroups from Figure 2 to the maximal `FID` marginal subgroup on the same feature. This is displayed as $|log_{10}(R)|$ where $R$ is the ratio of average importance per data point for separable notions and the ratio of coefficients for the linear coefficient notion. The feature associated with the subgroups is written above each bar.

## J  STATISTICAL VALIDITY OF RESULTS: GENERALIZATION OF `FID` AND $|g|$

When confirming the validity of our findings, there are two potential concerns: (1) Are the subgroup sizes found in-sample approximately the same on the test set and (2) do the `FID`'s found on the training set generalize out of sample? Taken together, (1) and (2) are sufficient to guarantee our maximal `AVG-SEPFID` values generalize out of sample.

In Figure 7, we can see that when we take the maximal subgroup found for each feature $f_j$, $g_j^*$, and compute it's size $|g_j^*|$ on the test set, for both the separable and non-separable methods it almost always fell within the specified $[\alpha_L, \alpha_U]$ range; the average difference in $|g_j^*(X_{train})|$ and $|g_j^*(X_{test})|$ was less than .005 on all notions of feature importance and all datasets except for Student, which was closer to .025 due to its smaller size. A few rare subgroups were significantly outside the desired $\alpha$ range, which was typically due to the degenerate case of the feature importance values all being 0 for the feature in question. Additional plots for all (dataset, notion) pairs are in Appendix L.

In Figure 8, we compare `AVG-SEPFID`$(f_j, g_j^*, X_{train})$ to `AVG-SEPFID`$(f_j, g_j^*, X_{test})$, or `LIN-FID` in the case of the linear regression notion, to see how `FID` generalizes. The separable notions all generalized very well, producing very similar `AVG-SEPFID` values for in and out of sample tests. The non-separable method still generalized, although not nearly as robustly, with

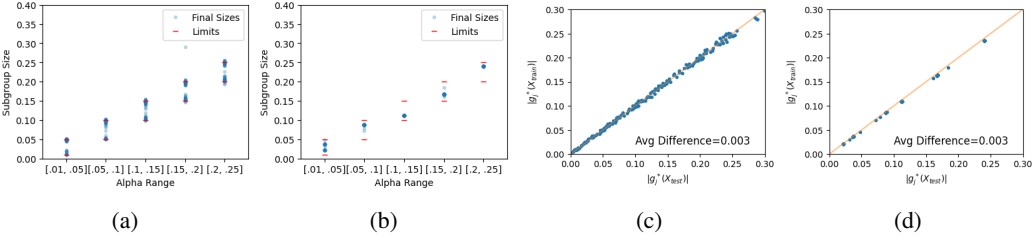

(a)           (b)           (c)           (d)

Figure 7: Generalizability of $|g|$ on the Folktables dataset. (a) Size outputs from Algorithm 1 for all features and separable notions and (b) from optimizing Equation 10 for LIN-FID show that our size constraints hold in-sample. (c) Plots the corresponding values of $|g_j^*(X_{train})|$ vs $|g_j^*(X_{test})|$ for separable notions and (d) for LIN-FID, showing that the subgroup size generalizes out of sample.

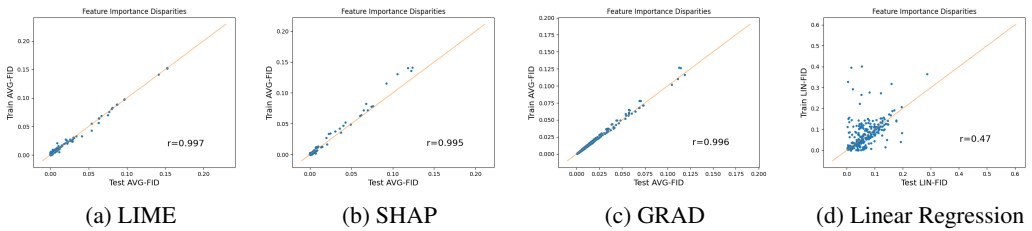

(a) LIME        (b) SHAP        (c) GRAD        (d) Linear Regression

Figure 8: Out of sample generalization of the methods. Each dot represents a feature, plotting FID on $X_{test}$ vs on $X_{train}$. All are computed on the Folktables dataset except (c) is computed on COMPAS R. The diagonal line represents perfect generalization and the Pearson correlation coefficient is displayed in figure. The non-separable approach suffers from the instability of the WLS method.

outlier values occurring. This was due to ill-conditioned design matrices for small subgroups leading to instability in fitting the least squares estimator. In Appendix O, we investigate the robustness of the feature importance notions, evaluated on the entire dataset. We find that the coefficients of linear regression are not as stable, indicating the lack of generalization in Figure 8 could be due to the feature importance notion itself lacking robustness, rather than an over-fit selection of $g_j^*$.

## K    CHOICE OF HYPOTHESIS CLASS

One ablation study we explored was the choice of classification model $h$. While the main experiments used a random forest model, we also explored using a logistic regression model. The logistic regression model was implemented with the default sklearn hyperparameters. We found that the results are roughly consistent with each other no matter the choice of $h$. In Table 5 and Table 6, we see that the features with the highest AVG-SEPFID, their subgroup sizes, and the AVG-SEPFID values are consistent between the choice of hypothesis class. We then looked further into the features that were used to define these subgroups. In Figure 9, we see that the subgroups with high AVG-SEPFID for the feature Age were both defined by young, non-Asians.

Similarly consistent results were found across all feature importance notions and datasets. As a result, all of the results presented in the main section of the paper used random forest as the hypothesis class.

## L    SUBGROUP SIZES OUTCOMES

In Figure 10 we chart the subgroup sizes, $|g(X_{test})|$, outputted by the algorithms across all dataset and importance notion combinations. As a whole, the final subgroup sizes were generally within the specified $\alpha$ range. Occasionally, there were subgroups which were significantly outside the expected range. Usually this was due to most of the importance values, $F(f_j, X, h)$, being zero for a given feature.

| | $h = $ Random Forest | | | $h = $ Logistic Regression | |
|---|---|---|---|---|---|
| Feature | Size | `AVG-SEPFID` | Feature | Size | `AVG-SEPFID` |
| Age | $.05 - .1$ | $.144$ | Age | $.05 - .1$ | $.21$ |
| Priors Count | $.01 - .05$ | $.089$ | Priors count | $.01 - .05$ | $.092$ |
| Juv Other Count | $.01 - .05$ | $.055$ | Juv Other Count | $.01 - .05$ | $.055$ |
| Other Features | - | $< .025$ | Other Features | - | $< .025$ |

Table 5: Comparing results between using random forest and logistic regression as the hypothesis class for classifier $h$ using LIME as the importance notion on the COMPAS R dataset. Here we display the features with the highest `AVG-SEPFID`, the subgroup size $|g|$, and the `AVG-SEPFID`. We can see that the choice of hypothesis class $h$ does not substantially affect the output. We used random forest for all of our main experiments.

| | $h = $ Random Forest | | | $h = $ Logistic Regression | |
|---|---|---|---|---|---|
| Feature | Size | `AVG-SEPFID` | Feature | Size | `AVG-SEPFID` |
| Age | $.01 - .05$ | $.4$ | Age | $.01 - .05$ | $.21$ |
| Priors Count | $.01 - .05$ | $.11$ | Priors count | $.01 - .05$ | $.14$ |
| Other Features | - | $< .05$ | Other Features | - | $< .05$ |

Table 6: Same as Table 5 except using SHAP as the importance notion. With SHAP, there were fewer features with significant `AVG-SEPFID` before dropping off but in both cases, the choice of $h$ did not significantly affect the outcome.

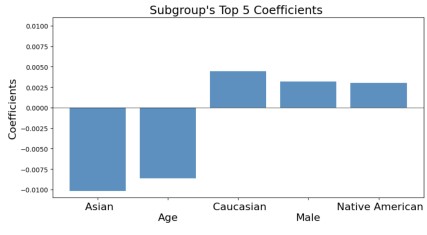
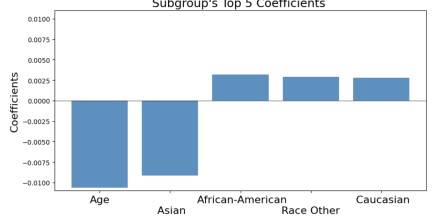

(a) $h = $ Random Forest, $f_{j*} = $ Age  (b) $h = $ Logistic Regression, $f_{j*} = $ Age

Figure 9: Comparing the choice of hypothesis class of $h$. Here we show the defining coefficients for the highest `AVG-SEPFID` subgroup found on the COMPAS R dataset using LIME as the feature importance notion. For the feature `Age`, we find that young and non-Asian were the two most defining coefficients for $g^*$ no matter which choice of $h$.

In Figure 11 we compare $|g_j^*(X_{train})|$ and $|g_j^*(X_{test})|$, outputted by the algorithms across all dataset and importance notion combinations. As we can see, the subgroup sizes were very consistent between the train and test set meaning $|g|$ generalized very well. The average difference was only somewhat large on the Student dataset, due to the fact that it is a smaller dataset.

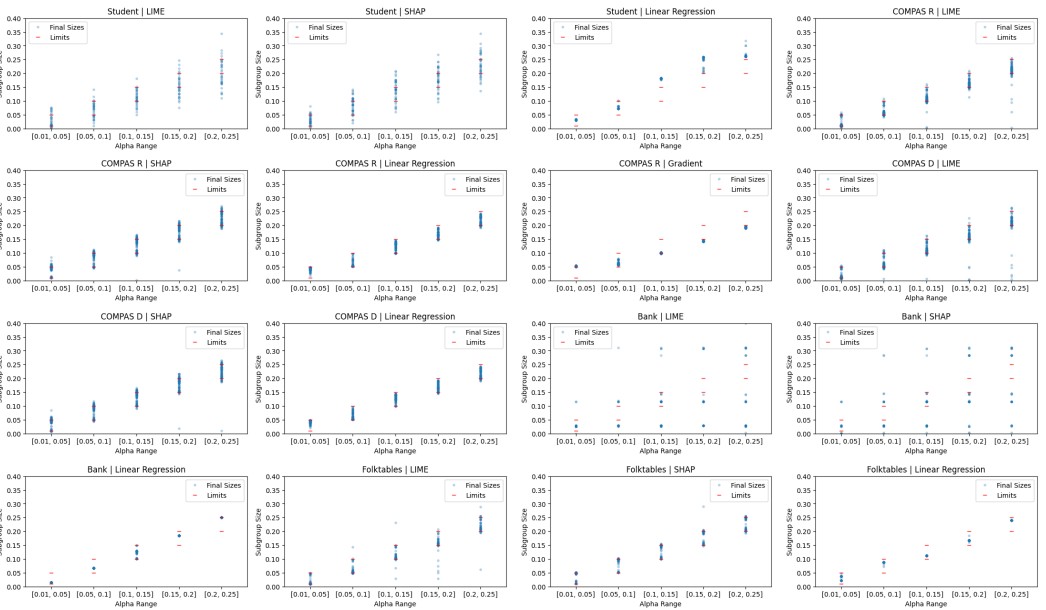

Figure 10: Final subgroup sizes of $g(X_{test})$ compared with $\alpha$ range. These almost always fall within the correct size range. Student has the largest errors, mostly due to the fact that the dataset itself is small.

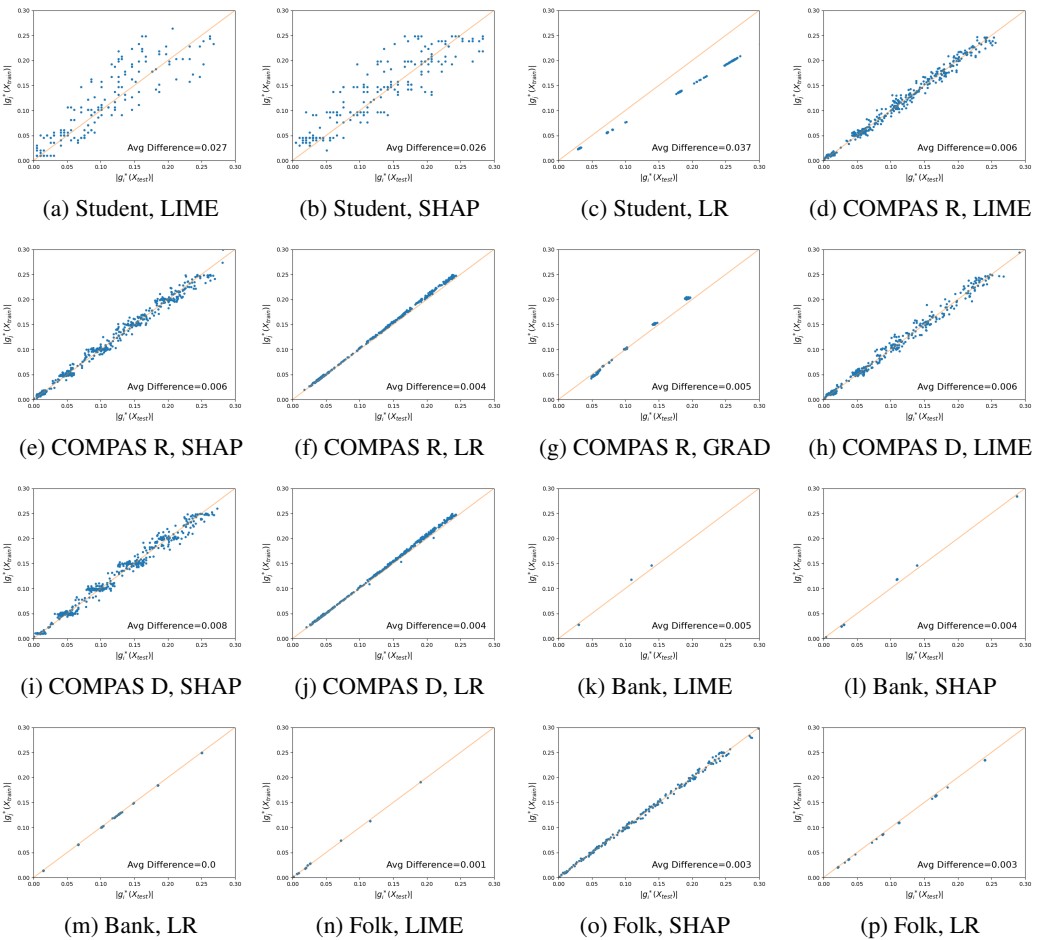

Figure 11: Comparing $|g_j^*(X_{train})|$ and $|g_j^*(X_{test})|$. We can see that the size of the subgroup was consistent between the train and test set.

# M    ALGORITHM 1 OPTIMIZATION CONVERGENCE

Here are additional graphs showing examples of the convergence of Algorithm 1. Data was tracked every 10 iterations, recording the Lagrangian values (to compute the error $v_t = max(|L(\hat{p}_{\mathcal{G}}^t, \hat{p}_{\lambda}^t) - \underline{L}|, |\overline{L} - L(\hat{p}_{\mathcal{G}}^t, \hat{p}_{\lambda}^t)|)$), the subgroup size, and AVG-SEPFID value, graphed respectively in Figure 12. We can see AVG-SEPFID value moving upward, except when the subgroup size is outside the $\alpha$ range, and the Lagrangian error converging upon the set error bound $v$ before terminating.

While Theorem 1 states that convergence time may grow quadratically, in practice we found that computation time was not a significant concern. The time for convergence varied slightly based on dataset but for the most part, convergence for a given feature was achieved in a handful of iterations that took a few seconds to compute. Features which took several thousand iterations could take around 30 minutes to compute on larger datasets.

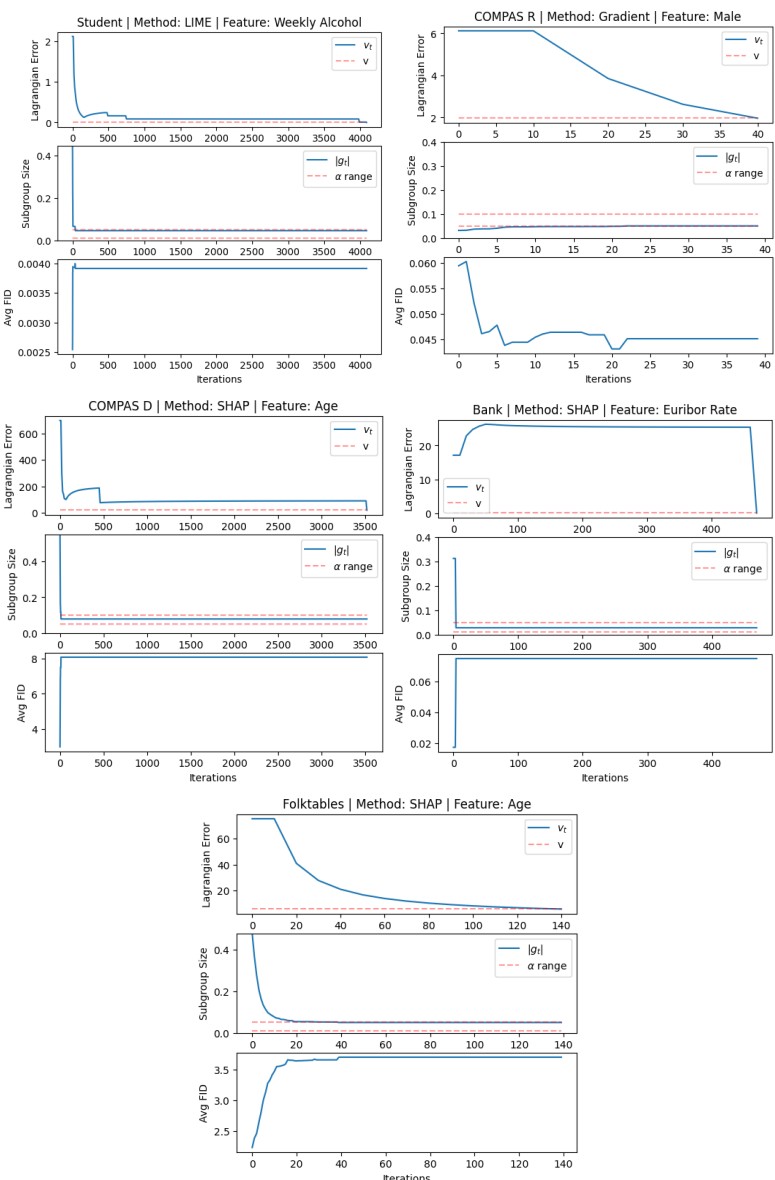

Figure 12: Plots detailing the convergence of Algorithm 1. The top plot shows the error convergence, i.e. the max difference in Lagrangian values between our solution and the min/max-players' solution. The other two plots display the subgroup size and AVG-SEPFID of the solution. Convergence almost always happened in fewer than 5000 iterations, allaying concerns about theoretical run time.

# N    NON-SEPARABLE OPTIMIZATION CONVERGENCE

Here are additional graphs showing the convergence in the non-separable approach. Using the loss function that rewards minimizing the linear regression coefficient (or maximizing it) and having a size within the alpha constraints, we typically reach convergence after a few hundred iterations. In Figure 13, we can see in the respective upper graphs that the subgroup size converges to the specified $\alpha$ range and stays there. Meanwhile, in the lower graph, we see the LIN-FID attempt to maximize but oscillates as the appropriate size is found.

Convergence using this method was almost always achieved in under 1000 iterations. Running this for all features took around 2 hours to compute on the largest datasets. The optimization was run using GPU computing on NVIDIA Tesla V100s.

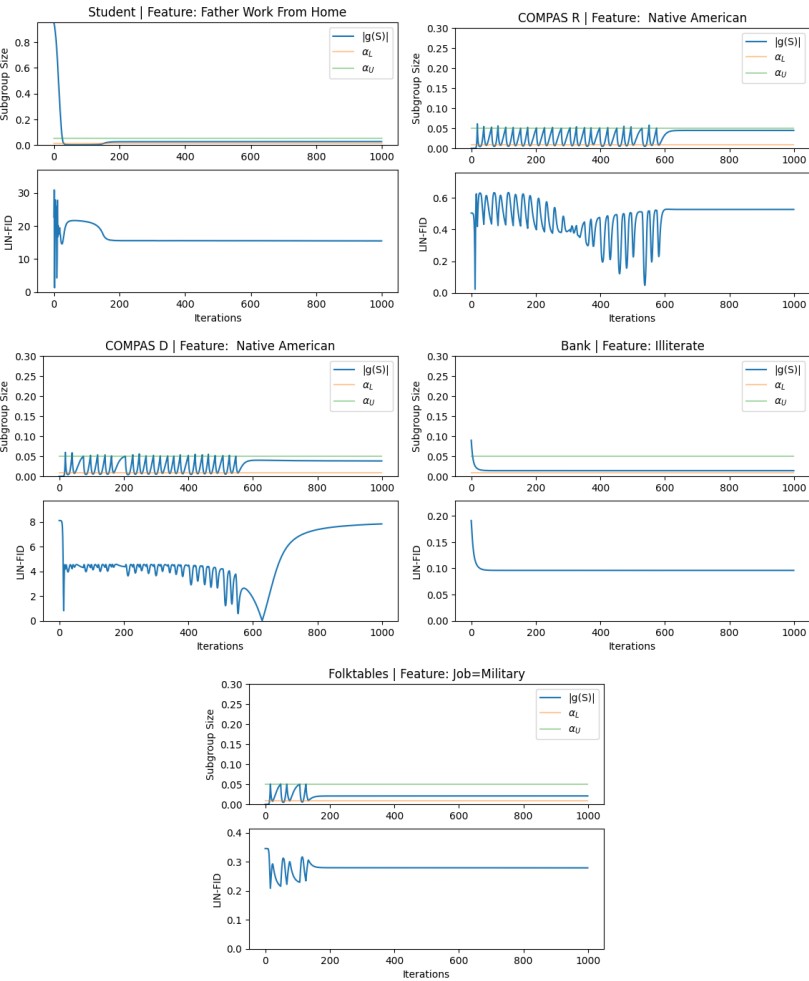

Figure 13: Plots of subgroup size and linear regression coefficient of $g$ over the training iterations of the Adam optimizer. For each dataset, the feature with the highest LIN-FID was displayed.

## O   IMPORTANCE NOTION CONSISTENCY

To see how consistent importance notion methods were, we plotted the values of $F(f_j, X_{test}, h)$ against $F(f_j, X_{train}, h)$ with each point representing a feature $f_j$ of the COMPAS dataset. The closer these points track the diagonal line, the more consistent a method is in providing the importance values. As we can see in Figure 14, LIME and GRAD are extremely consistent. Linear regression is less consistent, due to instability in fitting the least squares estimator on ill-conditioned design matrices. SHAP is also inconsistent in its feature importance attribution, however the `AVG-SEPFID` still generalized well as seen in Figure 8. This could mean that while SHAP is inconsistent from dataset to dataset, it is consistent relative to itself. i.e. if $F(j, X_{train}) > F(j, X_{test})$ then $F(j, g(X_{train})) > F(f, X_{test})$ meaning the `AVG-SEPFID` value would remain the same.

These inconsistencies seem to be inherent in some of these explainability methods as noted in other research Krishna et al. (2022); Dai et al. (2022); Agarwal et al. (2022a); Alvarez-Melis & Jaakkola (2018); Bansal et al. (2020). Exploring these generalization properties would be an exciting future direction for this work.

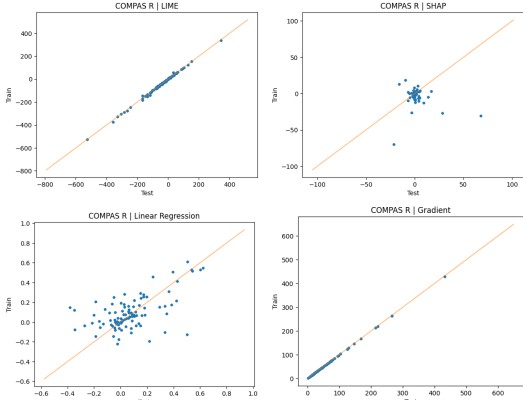

Figure 14: Consistencies of importance notions. Each point represents a feature, the x-value is $F(j, X_{test})$, and y-value is $F(j, X_{train})$. The closer the points are to the diagonal, the more consistent the notion is.

