# OpenReview forum: "Model Explanation Disparities as a Fairness Diagnostic"
_ICLR.cc/2024/Conference — Submitted to ICLR 2024_

### Official Review · Reviewer_nvHG · 2023-11-01

**Soundness:** 2 fair
**Presentation:** 2 fair
**Contribution:** 2 fair
**Rating:** 5
**Confidence:** 2

**Summary:**

In this work, the authors propose to leverage model interpretability methods to explore fairness problems. Specifically, based on local feature importance methods, they introduced a novel notion, FID (feature importance disparity), as a bias indicator under rich group fairness scenarios. Furthermore, they designed an oracle-efficient algorithm to search the large FID subgroups and claimed the potential utility of the proposed concept in fairness research.

**Strengths:**

- This paper is informative and well-written.

- Combining interpretability with the fairness problem, the idea is interesting.

**Weaknesses:**

- Not discussing specific definitions of fairness in the context of rich group fairness can sidestep a certain amount of risk, but does it affect the accuracy of the article and the related discussions?

- Section 3 and the proposed algorithm are a little confusing; please refer to the questions.

- The main body of the article exceeded the required number of pages.

**Questions:**

- I'm curious if the assumption about being locally separable is too strong.

- In algorithm 1, what is the definition of L?

- Are there any specific application scenarios or examples for the proposed FID metrics?

---

> ### Author Response · Authors · 2023-11-22
>
> $\textbf{Weakness: Fairness Discussion}$
>
> This is a great point and one that we looked to provide a more robust discussion of by the addition of two experiments in our new Section 4.3 connecting our work to existing fairness work. First, we used the open source GerryFair code by Kearns et al. 2018 to find rich subgroups that maximally violate false positive rate. We find that these rich subgroups also have high AVG-SEPFID, which in some cases are on the same features found using our methodology. These AVG-SEPFID values are lower than ones from our approach but the connection between the two highlights the relevance of our work towards fairness. We also evaluated traditional fairness metrics (independence, separation, and sufficiency as defined by Barocas et al. 2023) on our subgroups versus the whole dataset and found disparities along these metrics. These disparities were not always very large, but they did consistently appear. Given these two results, our work remains consistent with established literature on fairness.
>
> We note that regardless of the fairness metric used, establishing our results without optimizing for a specific fairness metric allows our method to be generalizable to all definitions of fairness. As discussed in Limitations: "It is known that even the most popular and natural fairness metrics are impossible to satisfy simultaneously, and so (choosing a fairness definition) would run up against the problem of determining what it means for a model to be fair." The results of our methodology may then be observed under any users' chosen perspective of fairness and still be accurate.
>
> $\textbf{Q: Local Separability Assumption}$
>
> We do not believe that a local separability assumption is too strong. The most popular model explanation methods such as the ones we use in our experiments (LIME, SHAP, and Vanilla Gradient) satisfy local separability. With rising interest in understanding individual model outputs, this will possibly continue to be the case. Furthermore, as seen in the experiments section, we also ran a successful experiment without the local separability condition, using linear regression coefficients as the importance notion. This shows that even if local separability is not satisfied, we can still identify subgroups with high AVG-SEPFID.
>
> $\textbf{Q: Section 3/Algorithm 1 Clarity}$
>
> Thank you for pointing this out. While we discuss this in full detail in Appendix B, we have added additional clarifying text to Section 3 to address this and other concerns about the presentation in this section. $L$ is the Lagrangian function that is formed in order to solve the constrained optimization problem in Equation 1. Because we are optimizing over randomized classifiers in $\Delta(\mathcal{G})$, we can apply strong duality to solve the dual problem:
>
>
> $(p_g^*, \lambda^*) = argmin_{p_g \in \Delta(\mathcal{G})} argmax_{\lambda \in \Lambda} \mathbb{E}_{g \sim p_g}[L(g, \lambda)]$
>
> $= argmin_{p_g \in \Delta(\mathcal{G})} argmax_{\lambda \in \Lambda}L(p_g, \lambda)$
>
> with $L(g, \lambda) = \sum_{x \in X} g(x) F(f_j,x,h) + \lambda_L \Phi_L + \lambda_U \Phi_U, \quad L(p_g, \lambda) = \mathbb{E}_{g \sim p_g}[L(g, \lambda)],$
>
> where $p^*_g$ is the optimal rich subgroup distribution, and $\lambda^*$ is the optimal value of the dual variable. $L(g,\lambda)$ as represented above is the sum of the total importance of feature $f_j$ on subgroup $g$ and the size constraint penalty functions $\Phi$ which are weighted by the dual $\lambda$ variables. This is the objective function that both the min and max player are trying to optimize when applying the result of Freund and Schapire 1996, each optimizing over different arguments.
>
> $\textbf{Q: Practical applications}$
>
> This is a good point and we have added additional discussion in the introduction along with the two substantial experiments mentioned above to address this. At a high level, our method is best viewed as an exploratory data analysis technique that can result in **hypothesis generation about potential sources of bias**. For example, if a given feature is found to have a significant effect in a given subgroup, this could result in the hypothesis that there is some measurement error in the way it is collected, or alternatively that the underlying dynamics are actually different for that subgroup, in which case a reasonable solution may be to fit a separate model for that subgroup.
>
> Our additional experiments highlight two other scenarios where our methodology may be useful. First, the FID metric may be used to audit a subgroup which we already know violates some fairness metric to aid a practitioner in quickly identifying features which may be the source of bias or error. Second, Algorithm 1 can be used to find rich subgroups that might face some kind of bias, as we showed that high FID subgroups tend to have other fairness metric disparities.

---

### Official Review · Reviewer_tQK6 · 2023-11-05

**Soundness:** 2 fair
**Presentation:** 3 good
**Contribution:** 2 fair
**Rating:** 3
**Confidence:** 4

**Summary:**

This paper introduces the notion of feature importance disparity (FID) to capture fairness of a model in terms of whether a feature importance value differs significantly for a subgroup compared to the entire population. Then to assess the fairness of a given model, the authors present an algorithm to find a feature and a subgroup, characterized by a function over the protected features, that exhibits a large FID. The algorithm involves iteratively solving constrained optimization problems using a polynomial number of calls to an oracle for cost-sensitive classification. The proposed method is evaluated empirically on four datasets.

**Strengths:**

The characterization of fairness in terms of disparities in feature importance scores is novel and interesting.

The proposed approach can be used with many feature importance notions and (somewhat) model-agnostic (for some feature importance scores), and thus can potentially be widely applicable.

The paper is well-written and easy to follow.

**Weaknesses:**

The proposed method returns real-valued subgroup functions (instead of binary) which are not really interpretable. Binary subgroup functions are indicators for some population groups that would be affected disparately. On the other hand, while real-valued subgroup functions can be interpreted as fractional membership in some demographic groups, their use in characterizing bias against/towards certain subgroups is questionable.

The number of calls to the CSC oracle required by Algorithm 1 to maximize AVG-SEPFID is quadratic in the size of data, for a single feature. This is quite computationally expensive, which would limit the applicability of the proposed method to large datasets with a large number of features.

Empirical evaluation is lacking comparison with any baseline. For example, how do the subgroups identified by the proposed method compare to existing `rich subgroup’ discovery methods?

**Questions:**

1. How do the subgroups identified by the proposed method compare to existing `rich subgroup’ discovery methods on the four datasets?

2. What were the runtimes for optimizing AVG-SEPFID for the experiments?

3. Given a subgroup (with fractional membership) and a feature with high FID, how do the authors suggest the model or domain expert address this? Moreover, how do you decide what value of AVG-SEPFID is significantly high to signal bias?

4. The approximation bound given by Theorem 1 seems quite weak. Even though the difference between the optimum and the expected FID over the distribution p_g is bounded, the expected difference for each g (E[FID(j,g*)-FID(j,g)]) may still be large. Is it possible to bound such expected difference or give a probabilistic guarantee?

Minor comments / questions:
- Definition 3: typo: $\frac{1}{n|g|}$ -> $\frac{1}{|g|}$
- What does bound B in Algorithm 1 represent intuitively?
- According to the experimental results regarding coefficients of subgroup functions, it appears that the subgroup functions g are linear functions. Is this a restriction imposed by the proposed algorithm?
- While the FID is useful in the algorithm to optimize for AVG-SEPFID, I don’t think it makes much sense as a fairness notion itself. According to Definitions 1 and 2, separable FID just ends up being $|\sum_X (g(x)-1) F(f_j,X,h)|$, which roughly corresponds to the sum of feature importance of non-group data points.

---

> ### Author Response · Authors · 2023-11-22
> **Addressing Main Questions**
>
> $\textbf{Q1: Comparison to existing rich subgroup work}$
>
> While there is existing literature exploring rich subgroups, we are the first to do work in the context of feature importance disparities. Prior rich subgroup discovery methods would not be applicable for optimizing for FID. We did, however, run an additional experiment in Section 4.3 using the GerryFair code of Kearns et al. 2018 to find rich subgroups that maximize false positive rates and audit them with respect to AVG-SEPFID. We find that these subgroups also have significant AVG-SEPFID on some features, but not as large as those found by our Algorithm 1 that optimizes for AVG-SEPFID. While this experiment was primarily run to address other reviewers' concerns about connections to fairness, it also highlights the connection between our work and rich subgroup discovery algorithms designed for fairness metrics like equalized odds.
>
> $\textbf{Q2: Runtime concerns}$
>
> This is a good question and a reasonable concern that we found in practice not to be an issue. As discussed in Appendix M: "While Theorem 1 states that convergence time may grow quadratically, in practice we found that computation time was not a significant concern -- empirically our method converged must faster than the theory alone would suggest. The time for convergence varied slightly based on dataset but for the most part, convergence for a given feature was achieved in a handful of iterations that took a few seconds to compute. Features which took several thousand iterations could take around 30 minutes to compute on larger datasets."
>
> $\textbf{Q3: Steps forward}$
>
> These are great questions and we have made additions to the paper's introduction that better address the usefulness of our approach. We've added the following text to the introduction:
>
> "As we later discuss in Section 5, disparities in feature importance do not necessarily imply that a subgroup has fairness disparity as measured by conventional metrics like equalized odds or calibration, although we show in Subsection 4.3 that this is empirically often the case. Nor does finding a subgroup with high feature disparity come with a pre-defined “fix”– the disparity could be caused by many factors including (i) true underlying differences between subgroups, or (ii) differences in measurement of the features or outcome variables across subgroups. Since the specific “fixes” are highly context dependent, **our method should be viewed as a tool for generating hypotheses about potential sources of bias**, which can then be addressed by other means. For example, in Figure 4 we find that the feature arrested-but-with-no-charges is highly important when predicting two-year-recidivism on the population as a whole, but carries almost no importance on a subgroup which is largely defined by Native-American males. This could motivate further research into if this subgroup is policed in a different way than the population as a whole. Alternatively, a technical solution may be to train a separate model for this subgroup."
>
> With regards to the second question, the exact values of AVG-SEPFID are dependent on the feature importance notion and the dataset. Thus a threshold value would not be appropriate for signaling concern. The better signal for concern would be the AVG-SEPFID value relative to the average feature importance value. We represent it like this in Figure 2, where this ratio value ranges from a small multiplicative factor to over $225$ times the average value. We also note that AVG-SEPFID is not itself a bias metric, rather it may help identify sources of bias as discussed earlier.
>
> $\textbf{Q4: Approximation bound in Theorem 1}$
>
> I think there is a slight misunderstanding here -- this is a stronger result than a high probability guarantee or an expected error guarantee; the theorem is that (assuming access to CSC oracles) we return an approximately optimal rich subgroup distribution in $O(4n^2B^2/\nu^2)$ iterations, with probability $1$. This follows from the fact that the regret bound for the exponentiated gradient algorithm is with probability $1$.
>
> $\textbf{Weakness: Interpretability of subgroup}$
>
> In our methodology we select the best subgroup out of the distribution generated. This subgroup is a threshold function so **the outcome subgroup is still binary.** From Section 3: "However, in practice we simply take the groups $g_t$ found at each round and output the ones that are in the appropriate size range, and have largest FID values. The results in Section 4 validate that this heuristic choice is able to find groups that are both feasible and have large FID values. This method also generalizes out of sample showing that the FID is not artificially inflated by multiple testing (Appendix J). Moreover, our method provides a menu of potential groups $(g_t)^T_{t=1}$ that can be quickly evaluated for large FID, which can be a useful feature to find interesting biases not present in the maximal subgroup."

---

> ### Author Response · Authors · 2023-11-22
> **Addressing Minor Comments**
>
> $\textbf{c: Definition 3 typo}$
>
> Thank you for pointing this out. We have corrected this typo.
>
> $\textbf{q: What does the bound B in Algorithm 1 represent intuitively?}$
>
> The bound constant $B$ is a hyperparameter that represents the magnitude of the cost of violating the size constraints. Intuitively, it can be thought of like a coefficient for a gradient function where using too small of a value means the algorithm will not converge to a subgroup of appropriate size while using too large of a value means the proposed subgroup will jump around too much between iterations. As discussed in Appendix G, we found an appropriate value to be $B = 10^4 \cdot \mu(f_j)$ where $\mu(f_j)$ is the average feature importance value.
>
> $\textbf{q:  Subgroup functions g are linear functions. Is this a restriction imposed by the proposed algorithm?}$
>
> We set the hypothesis class of $g$ to be linear but that is not a restriction of our proposed algorithm. Following Kearns et al 2018 we picked a linear threshold function because it is (i) interpretable, (ii) generalizes out of sample, and (iii) is still significantly more powerful than looking at marginal attributes alone. Hebert-Johnson et al. use rich subgroups defined by conjunctions of attributes which would also be a reasonable choice for our setting. The framework is flexible enough to accommodate any hypothesis class assuming we have access to an approximate optimization oracle for that class.
>
> $\textbf{q: FID usefulness}$
>
> You are correct that FID itself is not representative of the true importance a separable explanation model assigns because it depends on the size of the subgroup. **This is why we use AVG-SEPFID as the metric to optimize for and which we report all of our results for.** The presentation of Definition $1$ builds the foundation for optimizing AVG-SEPFID and is also useful for non-separable notions of importance which do not run into the issue you pointed out.
>
> On a broader scope, we do not view FID or AVG-SEPFID as fairness notions in and of themselves. As discussed in the primary response, the overarching use of this work is to audit models for rich subgroups with potential fairness concerns, which we expanded upon with additional discussion in the introduction and the addition of the experiments in Section 4.3. We did not pick a traditional fairness metric to optimize for in order to remain agnostic about the "correct" fairness metric to use. As discussed in the Limitations section: "Importantly, we eschew any broader claims that large FID values necessarily imply a mathematical conclusion about the fairness of the underlying classification model in all cases. It is known that even the most popular and natural fairness metrics are impossible to satisfy simultaneously, and so we would run up against the problem of determining what it means for a model to be fair."

---

### Official Review · Reviewer_SE4V · 2023-11-09

**Soundness:** 3 good
**Presentation:** 2 fair
**Contribution:** 2 fair
**Rating:** 5
**Confidence:** 2

**Summary:**

The paper explores feature importance methods and their potential for disparities in attributing feature importance values across different subgroups. The authors introduce the concept of Feature Importance Disparity (FID), which measures whether local feature importance methods attribute different feature importance values on average in protected subgroups compared to the entire population as a proxy to indicate bias in the model or data generation process. The paper designs an efficient algorithm to identify subgroups with large FID and provides empirical results across 4 public datasets and 4 common feature importance methods.

**Strengths:**

- The paper does a good job at motivating the connection between explainability and fairness and why more exploration in this direction is warranted
- The experiments to showcase the results in terms of explainability for FID are extensive
- The convergence proof for Algorithm 1 appears correct to me

**Weaknesses:**

For completeness, I do not have any experience in explainability, but I do in algorithmic fairness as well as the optimization literature.
The main weaknesses of this paper to me appear to be (a) the presentation of FID and Algorithm 1 and (b) the implications and connections to downstream fairness.

(a) The technical presentation of the optimization algorithm is too abrupt for the reader, and some lingering questions never get addresses fully.
-What is the rich subgroup class we are considering, for practical examples?
-What do the size violations correspond to?
-The notion of CSC being relegated to the appendix really subtract clarity from the work, as the reader struggles to bounce back and forth from the appendix for almost every step of Algorithm 1.
I appreciate the intricacies of presenting an algorithm such as this one, but one could maybe shorten the "Introduction" section to provide either a (i) simple example or (ii) dedicate some space to go through each of the single ingredients of the algorithm. Personally, I think this would go a long way towards improving the understanding of AVG-SEPFID and appreciate why solving it is tricky.

(b) The main drawback of this paper is the lack of connections with fairness metrics (such demographic parity or equalized odds) to connect the results in e.g., Table 1, with the disparities of the classifiers with respect to different subgroups. While pointing out biases in the original data is useful, FID seems to be applicable to a given group or family of classifiers $\mathcal{H}$, while in practice what modelers are interested in is to be able to analyse and find the pathways of discrimination in a given model. I believe that an extra set of experiments where the connection between natural fairness metrics and the proposed FID would make this paper meaningful and interesting to the community. As of now, from an algorithmic fairness standpoint, the paper ends too abruptly.

**Questions:**

Please see "Weaknesses" section above.

---

> ### Author Response · Authors · 2023-11-22
> **Clarity + Connection to Fairness**
>
> $\textbf{Q(a): Clarity of Algorithm 1 and additional questions on rich subgroup class and size violations}$
>
> Thank you for highlighting concerns about the presentation. We have added additional context to the presentation of the algorithm including clarification of the approach after Equation 2 and more description of algorithm parameters.
>
> We provide algorithmic details of the rich subgroup class in Appendix G. The rich subgroup hypothesis class is a linear threshold function, which we will state more clearly in the draft. This matches prior work on rich subgroup fairness and multicalibration, which use the same rich subgroup function. Note that our rich subgroup class contains the class of conjunctions as a subclass.
>
> The size violations correspond to the upper and lower bounds on the rich subgroup size. As discussed in Section $3$, we need to constrain the rich subgroup size to $[\alpha_L, \alpha_U]$ since we are solving the constrained FID optimization problem over a discretization of $[0,1]$ and then taking the subgroup with maximal AVG-SEPFID. The proof and details for this approach are discussed in Appendix C.
>
> $\textbf{Q(b): Connection to fairness metrics}$
>
> This is a great point and one that we have addressed by making significant additions to the paper via the additional experiments in the new Section 4.3. We ran two additional experiments to tie in our notion of AVG-SEPFID with established fairness work. First, we used the open source GerrFair code by Kearns et al. 2018 to find rich subgroups that maximally violate false positive rates. We find that these rich subgroups also have high AVG-SEPFID, which in many cases load on the same feature found using our methodology. These AVG-SEPFID values are lower than ones from our approach, but the connection between the two highlights the relevance of our work towards fairness. We also evaluated traditional fairness metrics (equalized odds, calibration) on our subgroups g versus the whole dataset and found disparities along these metrics. These disparities were not always very large, but they did consistently appear.
>
> We also added some discussion about the role of our method to the introduction:
>
> "As we later discuss in Section 5, disparities in feature importance do not necessarily imply that a
> subgroup has fairness disparity as measured by conventional metrics like equalized odds or calibration,
> although we show in Subsection 4.3 that this is empirically often the case. Nor does finding a
> subgroup with high feature disparity come with a pre-defined “fix”– the disparity could be caused
> by many factors including (i) true underlying differences between subgroups, or (ii) differences in
> measurement of the features or outcome variables across subgroups. Since the specific “fixes” are
> highly context dependent, our method should be viewed as a tool for generating hypotheses about
> potential sources of bias, which can then be addressed by other means. For example, in Figure 4 we
> find that the feature arrested-but-with-no-charges is highly important when predicting
> two-year-recidivism on the population as a whole, but carries almost no importance on a
> subgroup which is largely defined by Native-American males. This could motivate further research
> into if this subgroup is policed in a different way than the population as a whole. Alternatively, a
> technical solution may be to train a separate model for this subgroup."
>
> Finally we note in the Limitations section: "Importantly, we eschew any broader claims that large FID values necessarily imply a mathematical conclusion about the fairness of the underlying classification model in all cases. It is known that even the most popular and natural fairness metrics are impossible to satisfy simultaneously, and so we would run up against the problem of determining what it means for a model to be fair." We still stand behind not using a fairness metric as our primary result measure, but we believe these two additional experiments added in the new Section 4.3 highlight the connection of our work to the broader fairness literature.

---

### Official Review · Reviewer_ZZsY · 2023-11-10

**Soundness:** 3 good
**Presentation:** 3 good
**Contribution:** 3 good
**Rating:** 6
**Confidence:** 3

**Summary:**

This paper formally introduces the notion of feature importance disparity in the context of ``rich'' subgroups, which could be used as a potential indicator of bias in the model/data generation process. The proposed algorithm finds (feature, subgroup) pairs that: (i) have subgroup feature importance that is often an order of magnitude different than the importance on the whole dataset, generalize out of sample.

**Strengths:**

The paper tackles an important question in machine learning literature on understanding how a specific feature contributes to a model's prediction, focusing on feature importance. The objective is clearly stated, definitions are well defined, and the optimization problem is intuitive and well-defined. Overall it uses existing methods in learning theory to solve an interesting fairness problem.

**Weaknesses:**

I find algorithm 1 hard to interpret; in particular, it involves in exponential gradient update, but it's not clear to me what certain parameters mean. In addition, it is not clear how this algorithm is related to Kivinen & Warmuth in terms of who is the max player, who is the min player, and what their objectives are.

**Questions:**

1. In definition 1, what does the expectation over? namely, what is $X$?
2. How should I intuitively understand the notion of ``Locally Separable''? In reality, how easy or hard is it to satisfy this condition?

Others:
1. The citation style is weird; the author might consider changing it.

---

> ### Author Response · Authors · 2023-11-22
>
> $\textbf{Q: Algorithm 1 Clarity}$
>
> Thank you for pointing this out, we have added some clarifying text to the description under Equation $2$ that improves the interpretability of algorithm 1. Regarding the relation to Kivinen and Warmuth, in short, to solve the constrained optimization problem in (1), we first form the Lagrangian and apply strong duality (which we can do because we are solving an LP after relaxing to optimizing over randomized classifiers in $\Delta(\mathcal{G})$), which corresponds to solving the dual problem:
>
> $(p_g^*, \lambda^*) = argmin_{p_g \in \Delta(\mathcal{G})} argmax_{\lambda \in \Lambda} \mathbb{E}_{g \sim p_g}[L(g, \lambda)]$
>
> $= argmin_{p_g \in \Delta(\mathcal{G})} argmax_{\lambda \in \Lambda}L(p_g, \lambda)$
>
> with $L(g, \lambda) = \sum_{x \in X} g(x) F(f_j,x,h) + \lambda_L \Phi_L + \lambda_U \Phi_U, \quad L(p_g, \lambda) = \mathbb{E}_{g \sim p_g}[L(g, \lambda)],$
>
> where $p^*_g$ is the optimal rich subgroup distribution, and $\lambda^*$ is the optimal value of the dual variable. The $\textit{min}$-player corresponds to optimizing over $p_g$ which aims to maximize the subgroup disparity (minimize the negative disparity), and the $\textit{max}$-player corresponds to optimizing $\lambda$, which is the vector of dual variables, one for each constraint. Both players have the same objective function $L(g, \lambda)$ since it is a zero sum game, but they are optimizing different arguments. We discuss this in full detail in Appendix B.
>
> $\textbf{Q: What is expectation over in Definition 1?}$
>
> The expectation is over $X$ which is our dataset of points drawn i.i.d. from some distribution $\mathcal{R}$. $X$ and $\mathcal{R}$ are both defined earlier in the preliminaries.
>
> $\textbf{Q: Locally Separable intuition}$
>
> Intuitively, if the underlying mechanism to compute an explanation value can be used on an individual data point, then the feature importance notion is locally separable. For example, LIME takes a single data point, computes a local model around it, and creates explanation values from that. On the other hand, coefficients from linear regression are often used by practitioners to explain feature importance but a linear regression may only be computed over a group of data points. In general, most modern popular model explanation methods are locally separable since current explainability work that can be applied to more complex models focuses on understanding individual predictions.

---

### Official Review · Reviewer_5Fo2 · 2023-11-10

**Soundness:** 3 good
**Presentation:** 3 good
**Contribution:** 3 good
**Rating:** 8
**Confidence:** 4

**Summary:**

The work introduces the problem of quantifying disparities in the output of a feature importance method across groups in the data. It finds subgroups for which the average feature importance for a given feature differs significantly compared to the feature importance in the whole population. The main contribution is that the subgroups need not be enumerated beforehand. The proposed method is able to search for groups represented as functions on an arbitrary set of features, called ‘rich subgroups’ in the fairness literature. To do so, the work formalizes a constrained optimization problem and solves it via online learning algorithms through a reduction that leverages oracle access to cost-sensitive classifiers. Theoretical result shows the solution is close to the maximum disparity group. Experiments on multiple datasets show the applicability of the method for different feature importance methods.

---
Updating the score after the rebuttal which addresses concerns on significance of the problem and presentation of the algorithm.

**Strengths:**

1. The ideas are presented clearly with adequate explanation and clear notation.
2. I like the generality of the problem formulation and how it is presented in Section 2 and 3 which can then be readily made specific to different feature importances.
3. Experiments are thorough - multiple explanation methods, datasets, tests on hold-out set - and are presented concisely.

**Weaknesses:**

1. (Major) Utility of the algorithm outputs could be discussed more thoroughly. As discussed in limitations, all differences in explanation outputs need not imply a discriminatory model. On the contrary, sometimes a model will differ in its logic in the protected group to account for differences in data. For example, the feature of not having a health insurance will be much more predictive of health outcomes in the unhoused population (say, a protected group) in comparison to whole population. So, the model logic and the feature importance will differ in the protected group. Reasons for finding such groups should then be discussed to motivate the utility of the method, along with suggestions for diagnosing the disparities. Introduction and empirical results can be edited to convey the significance of the explanation disparity problem.
2. (Minor) Algorithm for finding maximal group can be described in more detail. Reduction to CSC and the specifics of Algorithm 1 like what is theta, lambda, and so on can be briefly explained, even if this is a standard method for solving constrained optimization.

**Questions:**

1. Please discuss relation to the work Balagopalan et al. 2022 (The Road to Explainability is Paved with Bias: Measuring the Fairness of Explanations https://dl.acm.org/doi/abs/10.1145/3531146.3533179) which also investigates disparities in explanation outputs. Advantages of the proposed method such as rich subgroups should be highlighted.
2. Please include more discussion on what can be done after observing disparities in explanation output. How can the user interpret the relation between explanation disparities and the underlying social biases, and decide how to update the model?

---
## Minor (no response is requested)

In Definition 1 of FID, consider denoting if the expectation is over different samples of the whole dataset X^n.

The scope of prediction models, sensitive features, and feature importances supported by the method was not clear to me. Describe the assumptions needed on the prediction model (for CSC to give good solution) and sensitive features (mix of categorical, real variables) if any. Consider listing feature importance methods that satisfy separability and any other required conditions for the method, perhaps in a table format.

Consider discussing extensions to non-separable importances other than weights in linear regression such as feature-permutation based importances in decision trees.

Please increase font size in Figures 2-6.

---

> ### Author Response · Authors · 2023-11-22
> **Additional Experiments Addressing Relationship to Fairness Metrics + More**
>
> $\textbf{Q1: Discussion of Balagopalan}$
>
> Thank you for highlighting this paper, we have added discussion of it to the related works section. Balagopalan et al. 2022 is similar to Dai et al. 2022, which we already discuss. Those two papers investigate sensitive subgroups and their relationship with model explanation quality, as measured by fidelity, stability, and other metrics they define, whereas we look at the difference in the magnitude of the explanations. Furthermore, we generalize the problem significantly by considering rich subgroups and our algorithm for searching an exponentially large subgroup space is novel and is essential for any fairness work regarding rich subgroups. We have highlighted these differences in the related works discussion but these papers are certainly relevant and could be explored as a future work in conjunction with ours.
>
> $\textbf{Q2/Weakness 1: Model utility and next steps for user}$
>
> This is a great point -- we have both added significant discussion of this point in the introduction, and substantial experiments to make this more clear. At a high level, our method is best viewed as an exploratory data analysis technique that can result in hypothesis generation about potential sources of bias. For example, if a given feature is found to have a significant effect in a given subgroup, this could result in the hypothesis that there is some measurement error in the way it is collected, or alternatively that the underlying dynamics are actually different for that subgroup, in which case a reasonable solution may be to fit a separate model for that subgroup. While our method does not prescribe a given solution since it is very context dependent, it is a method to uncover these possible disparities in the first place. We've added the follow text to the introduction:
>
> "As we later discuss in Section 5, disparities in feature importance do not necessarily imply that a
> subgroup has fairness disparity as measured by conventional metrics like equalized odds or calibration,
> although we show in Subsection 4.3 that this is empirically often the case. Nor does finding a
> subgroup with high feature disparity come with a pre-defined “fix”– the disparity could be caused
> by many factors including (i) true underlying differences between subgroups, or (ii) differences in
> measurement of the features or outcome variables across subgroups. Since the specific “fixes” are
> highly context dependent, our method should be viewed as a tool for generating hypotheses about
> potential sources of bias, which can then be addressed by other means. For example, in Figure 4 we
> find that the feature arrested-but-with-no-charges is highly important when predicting
> two-year-recidivism on the population as a whole, but carries almost no importance on a
> subgroup which is largely defined by Native-American males. This could motivate further research
> into if this subgroup is policed in a different way than the population as a whole. Alternatively, a
> technical solution may be to train a separate model for this subgroup."
>
> $\textbf{Additional Experiments}.$
> We also run new experiments investigating the relationship between our notion of subgroup disparity and fairness metrics like subgroup calibration and equalized odds, which summarize in a new Section 4.3 in the updated draft. We find that across all data sets (i) our found subgroups often also correspond to subgroups with significant disparities in fairness metrics but (ii) consistent with the theory in fairness, which says that even common fairness metrics can not be simultaneously satisfied, we also find some subgroups that don't have a large fairness disparity. We also run the reverse experiment, taking rich subgroups that maximize the false positive rate disparity using the GerryFair code from Kearns et al. 2018, and auditing these rich subgroups with respect to AVG-SEPFID. We find that the AVG-SEPFID of these subgroups is significant, but not as large as those found by our Algorithm $1$ that explicitly optimizes the AVG-SEPFID.
>
> $\textbf{Additional Comments:}$
>
> We have added text to Section 3 of the paper that adds clarity to the presentation of Algorithm 1. We have made additional minor changes to the submission to address concerns about readability.

---

### Author Response · Authors · 2023-11-22
**High Level Rebuttal Summary**

We thank the reviewers for their helpful comments and have responded to each review separately below. There were two common critiques that we have addressed with significant revisions to the submission.

- First, there were questions about the relationship of our work to traditional fairness metrics and practical use cases. We have addressed this with additional text in the intro as well as the addition of two substantial experiments in the new Section 4.3. These experiments go a long way in bolstering the practical importance of our methodology and answering the questions provided by the reviewers.
- Second, there was a request for more clarity of Algorithm 1 and related variables which we have addressed with more detail provided in Section 3 and labeling in the Algorithm 1 write up. There were other also comments around additional related works as well as minor changes to improve readability that we have added to our revised submission.

To make space for the above changes, we have moved the old Section 4.3 on statistical validity to the Appendix along with other minor edits that do not affect the paper. We believe we have addressed the concerns thoroughly enough to merit an increased score, and note that none of the reviews found significant flaws in our methods and results.

---

### Meta-Review · Area_Chair_YbBM · 2023-12-09

**Metareview:**

The paper addresses the issue of quantifying disparities in feature importance across groups in a dataset. It identifies subgroups where the average feature importance for a specific feature significantly differs from the overall population, and notably, these subgroups do not need to be predefined. The method is designed to search for “rich subgroups” represented as functions on arbitrary feature sets. The approach formulates a constrained optimization problem and employs online learning algorithms to solve it through a reduction leveraging oracle access to cost-sensitive classifiers. Theoretical results demonstrate the proximity of the solution to the maximum disparity group. Empirical experiments on multiple datasets showcase the method's applicability to various feature importance methods, introducing the novel notion of feature importance disparity (FID) to assess model fairness.

Reviewers all agree this paper motivates strongly. But on the other hand, the paper’s clarity and impact, as of its current version, is hindered by missing of details and explanations. The reviewers identified a number of places that the authors can clarify with either more details or an accessible example. The authors are also encouraged to add more discussion on the connection between the proposed study and the existing fairness metrics.

**Justification For Why Not Higher Score:**

The paper has potential but would need a revision to clarify a number of technical details. The authors are also encouraged to better relate their work to the existing fairness metrics & treatments.

**Justification For Why Not Lower Score:**

N/A

---

### Decision · Program_Chairs · 2024-01-16

Reject